# Data-driven identification of predictive risk biomarkers for subgroups of osteoarthritis using interpretable machine learning

Rikke Linnemann Nielsen [1], Thomas Monfeuga [1], Robert R. Kitchen[1], Line Egerod [1], Luis G. Leal [1], August Thomas Hjortshøj Schreyer[1], Frederik Steensgaard Gade [2], Carol Sun [1], Marianne Helenius [3], Lotte Simonsen [2], Marianne Willert[2], Abd A. Tahrani [4], Zahra McVey[1] & Ramneek Gupta [1]✉

Osteoarthritis (OA) is increasing in prevalence and has a severe impact on patients' lives. However, our understanding of biomarkers driving OA risk remains limited. We developed a model predicting the five-year risk of OA diagnosis, integrating retrospective clinical, lifestyle and biomarker data from the UK Biobank (19,120 patients with OA, ROC-AUC: 0.72, 95%CI (0.71–0.73)). Higher age, BMI and prescription of non-steroidal anti-inflammatory drugs contributed most to increased OA risk prediction ahead of diagnosis. We identified 14 subgroups of OA risk profiles. These subgroups were validated in an independent set of patients evaluating the 11-year OA risk, with 88% of patients being uniquely assigned to one of the 14 subgroups. Individual OA risk profiles were characterised by personalised biomarkers. Omics integration demonstrated the predictive importance of key OA genes and pathways (e.g., *GDF5* and TGF-β signalling) and OA-specific biomarkers (e.g., CRTAC1 and COL9A1). In summary, this work identifies opportunities for personalised OA prevention and insights into its underlying pathogenesis.

Osteoarthritis (OA) is a common chronic degenerative joint disease, with an estimated 528 million people living with OA. The global prevalence of OA has increased by 48% from 1990 to 2019[1], and expected to increase further, due to ageing populations and the rise in obesity rates[1,2]. In addition to its health burden, OA has a high impact on health care expenditure and social care cost[3], with the economic impact ranging from 1 to 2.5% of gross national product (GNP) in some countries. The average annual cost of OA for an individual is estimated to be between $700–$15,600 (USD, 2019) across countries in Asia, Europe, North America and Oceania[1]. In addition, there are currently no approved curative treatments or therapies that impact disease progression. Patients are often diagnosed with late-stage disease[4], where the main treatment option is joint replacement surgery[5]. Hence, there is large interest and an unmet need to develop tools that can aid

early diagnosis and identify effective preventative, and disease modifying strategies.

It is crucial to improve our understanding of OA pathogenesis and develop appropriate prediction strategies to address the unmet need[5,6]. This is challenging due to the complexity of OA and its heterogeneity spanning multiple biological mechanisms and disease phenotypes[5,6]. It has been proposed that it would be beneficial to use data-driven and machine learning approaches for patient-specific prediction models in OA, to dissect the complex relationship between risk biomarkers[4,7–9]. There is also a gap in the use of machine learning methods for the prediction of OA across multiple joints[9]. Previous attempts of OA prediction models had several limitations such as a focus on knee OA, small sample sizes and restricted sets of input features[4,8,9]. Most of these studies have lacked comprehensive

[1]Novo Nordisk Research Centre Oxford, Oxford, UK. [2]Novo Nordisk A/S, Måløv, Denmark. [3]Technical University of Denmark, Kgs. Lyngby, Denmark. [4]Novo Nordisk A/S, Søborg, Denmark. ✉e-mail: RMGP@novonordisk.com

incorporation of genetics, clinical biomarkers and other environmental factors[4,8,10], and the assessment of the impact of changes in modifiable risk biomarkers[9]. All of these elements are needed to guide preventative strategies and precision medicine in OA. Additionally, multiple studies focused on the prediction of disease progression rather than prediction of disease incidence or diagnosis[4].

In this retrospective study, we develop a machine learning model to predict individual risk and identify risk biomarkers up to 5-years prior to an OA diagnosis. Through the integration of multi-modal patient data, we identify subgroups of OA, with different risk biomarker profiles, which is validated to be effective on an unseen subpopulation of the UK Biobank up to 11 years ahead of diagnosis. The model captures the broad risk biomarker landscape, in a UK cohort of ~20,000 people diagnosed with OA, utilising electronic health records (EHR), clinical biomarkers, self-reported questionnaire data, genomics, proteomics, and metabolomics on available subsets of individuals. The model quantifies the impact of risk biomarkers on the predicted OA risk at the population and individual level, enabling detailed estimation of the contribution of these biomarkers for OA risk.

## Results

### OA study population

The UK Biobank is a population-based cohort study with health information from assessments at the time of recruitment (2006–2010) and linkage to electronic health records (EHR) of individuals in the UK ($N = 502,476$)[11]. We identified 103,086 patients with an OA diagnosis from EHR data (~21% of all UK Biobank participants, Supplementary Fig. 1). In total, 55,628 OA diagnoses were identified from primary care settings (general practices, follow-up until 09/2017) and 49,318 OA diagnoses from secondary care settings (hospital inpatient data, follow-up until 03/2017). Clinical codes of OA diagnoses are given in Supplementary Data 1 (primary care: Read v2 and CTV3/Read v3, secondary care: ICD-9 or ICD-10).

Primary healthcare data is available for ~45% of the UK Biobank cohort which enabled capture of longitudinal data for a subset of patients that were diagnosed with OA ($N = 67,772$). An equal number of control participants who were never diagnosed with OA in the available EHR study period were identified ($N = 67,772$). Controls were randomly selected and date-matched with the OA diagnosis dates for case patients. Cases and controls were then filtered for those with an OA diagnosis/matched index date a maximum of 5 years after the UK Biobank recruitment assessment centre. We focussed our study on the diagnosis of OA up to 5 years after the assessment centre. This was to capture the risk biomarkers that are predictive of OA diagnosis in the focused period of 5 years prior to diagnosis, when patients are at high-risk, and a potential window to explore for preventative interventions with the deep phenotyping of the aging population. Controls were required to have observational data and no death registered during the 5 years prior to the index date (study period: 06/2006–09/2015). This resulted in a total of 19,120 patients with diagnosed OA and 19,252 controls included in the analysis (Fig. 1A, Supplementary Fig. 1). For the patients with OA, the specific joints affected were mapped to the following joint categories: foot (4%), spine (10%), arm (11%), hip (13%), and knee (28%), albeit most OA diagnoses were in unspecified joints (Fig. 1B).

In addition, an independent hold-out validation population was generated following the same procedure as above, but with a longer time between data collection at the assessment centre and OA diagnosis (5 years to 11 years). (Fig. 1A, study period: 09/2011–09/2017). Furthermore, samples with missing data for variables used in the clustering rules were excluded (non-imputed data used). This resulted in a population of 7341 cases and 5999 controls.

Comparison of baseline distribution for known OA risk biomarkers showed that OA cases in general were of older age, with higher BMI and a higher female-to-male ratio in comparison to controls in

both the OA study population and hold-out validation population (Fig. 1C).

### Risk modelling

Following identification of the OA study and validation populations, diverse multi-modal longitudinal patient data were processed for integration into an eXtreme Gradient Boosting (XGBoost) machine learning model (Fig. 2A, B). The XGBoost model was trained to predict the 5-year risk of OA diagnosis from retrospective data. The XGBoost model integrated clinical, sociodemographic, diet, physical activity, and lifestyle data from the recruitment assessment centre, with clinical data from the 5-year longitudinal EHR data extracted from available data ahead of OA diagnosis or matched index date (Clin model, Fig. 1A). The EHR data captured diagnosis of previous post-traumatic OA diagnosis, longitudinal blood and urine biomarkers, clinical measurements, as well as medication data for obesity, OA, and type 2 diabetes. Longitudinal data was captured in the 5 years prior to OA diagnosis/index date using yearly data bins (Supplementary Fig. 2). Missingness estimation for each feature included in the machine learning models is presented in Supplementary Data 2. Furthermore, diverse omics data were integrated into separate models for individuals where this information was available (genetics (ClinSNP, ClinWGPRS, ClinGRS, ClinPath), metabolomics (ClinMet), and proteomics (ClinPro), Fig. 2A). The interpretable machine learning framework was used to explore and quantify risk biomarkers of OA at population, precision, and personalised levels (Fig. 2C).

### Prediction of OA from 5-year multi-modal clinical data

Retrospective longitudinal clinical data were integrated in a XGBoost model to predict the 5-year risk of OA diagnosis (Clin model). Performance was evaluated in the test set in a 5*5 cross-validation (Fig. 2B). The Clin model achieved a cross-validated ROC-AUC performance of 0.72 (95%CI: 0.71–0.73, Fig. 3A–C). The Clin model was able to predict 7 in 10 patients who developed OA, and out of all the OA case predictions made by the model, 66% of these were true-positive OA cases (Fig. 3C). Conversely, we were able to predict 6 in 10 individuals who did not develop OA, with 67% of the predicted controls being true-negative controls (Fig. 3C). The Clin model's predictive performance was robust across random model initialisations and performed significantly better than models trained on permuted OA status labels (Supplementary Table 1). We assessed if the Clin model had a stronger predictive performance when predicting specific subgroups of OA across different affected joints, specified for some of the OA diagnoses, including arm, foot, hip, knee, or spine. Performance ranged with ROC-AUC: 0.67–0.73 (Fig. 3A) and was highest for weight-bearing joints (ROC-AUC: 0.73 and 0.72 for knee and hip OA prediction, respectively). However, performance for joint-stratified models was only modestly different when compared to the overall Clin model.

A baseline machine learning model predicting the 5-year risk of OA, using only age, sex, and BMI, that are well-known OA risk biomarkers, predicted OA with ROC-AUC: 0.67 (95%CI: 0.67–0.68). Comparison of the predicted risk probabilities at the individual level showed that the additional features in the Clin model compared to the simpler model (age, sex and BMI) significantly increased the confidence in individual risk predictions (Supplementary Fig. 3) in addition to increasing predictive performance.

### Population risk biomarkers predictive of OA risk

To interpret which risk biomarkers were most important for the prediction of an OA diagnosis, Shapley additive explanations (SHAP) values were calculated. SHAP values estimate the marginal conditional impact of a feature on the model's prediction in relation to all other features included in the model. The top three predictors of increased OA risk included higher age, prescription of non-steroidal anti-inflammatory drugs (NSAIDs) during the year prior to OA diagnosis,

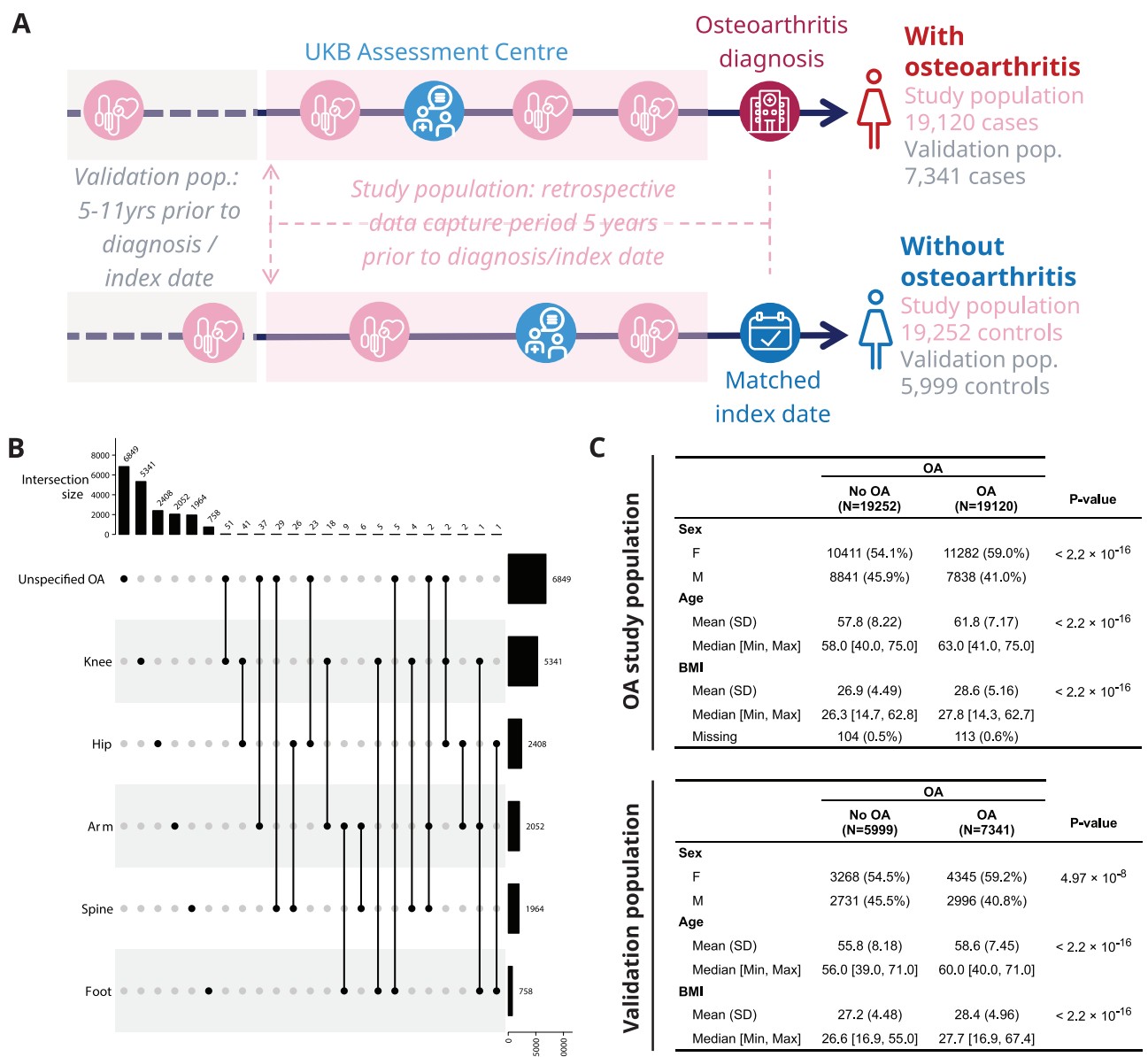

**Fig. 1 | Study design and population characteristics. A** Overview of study design including example of date matching cases and controls (longitudinal patient data). For patients (cases) diagnosed with osteoarthritis (OA), the OA diagnosis date was identified and a data capture period of 5 years prior to diagnosis created. For individuals not diagnosed with OA (controls), a matched index date, equivalent to the OA diagnosis date for the case used for matching, was identified. For controls, a data capture period of 5 years prior to the index date was created. For both cases and controls, longitudinal electronic health record (EHR) data and data from the UK Biobank assessment centre were captured in the 5-year data capture period. **B** Upset plot of joints affected that could be extracted from the OA diagnosis codes.

It was not possible to map all OA diagnoses to a specific joint (marked as unspecified OA). These groups were used for stratification in the prediction models. The set size represents the full number of patients that could be identified. When a patient with OA had multiple joints affected, both diagnoses were included in this joint mapping and hence the set size reflects the 19,120 OA cases identified. The intersection sizes represent the number of patients with at least the given set of diagnoses. **C** OA risk factors summarised across OA cases (OA) and matched non-OA controls (No OA) used for modelling in the study. *P*-values (uncorrected for multiple testing) were generated by two-sided Welch's *t*-test for continuous features and chi-squared test of independence for sex (F = Female, M = Male).

and higher BMI compared to individuals that did not develop OA (Fig. 3D). Following these three risk biomarkers, predictive contributions were made by a variety of features across individuals. Participants who rated their own health as excellent and had a faster walking pace, had a lower risk of OA. Higher levels of vitamin D were predictive of increased OA risk. A post-hoc analysis showed that individuals that reported taking vitamin D supplements typically had higher levels of vitamin D, a higher age and was more common in women with menopause which might contribute to the observed association between higher vitamin D levels and OA risk (Supplementary Fig. 4A–C respectively). Additionally, a higher hand grip strength, and a lower

ratio of fat mass to fat free mass were also predictive of lower OA risk. Individuals that had a higher socioeconomic status, as indicated by having a college or university degree and higher income, had a lower risk of OA. In contrast, people working heavy manual or physical jobs, or doing shift work, had increased risk of OA.

**Precision subgroups of OA**

The Clin model confirmed that the biological and environmental risk factors underlying OA are heterogenous across individuals. We attempted to capture this heterogeneity and categorise patients into subgroups with differing risk biomarker profiles. Hence, we clustered

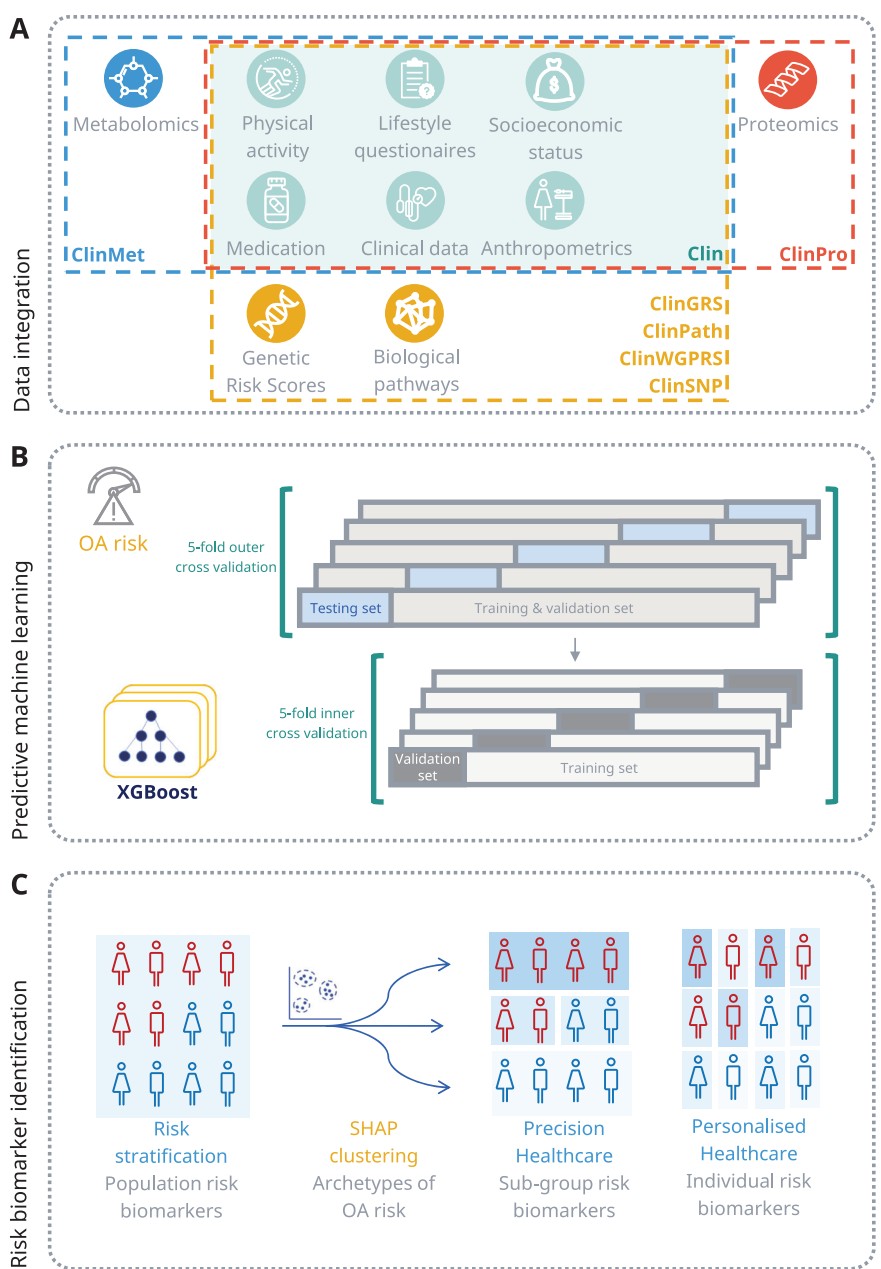

**Fig. 2 | Identification of OA risk biomarkers using machine learning. A** Data integration strategy overview for multi-omics data. **B** Study design for predictive machine learning modelling setup for training and model validation. **C** Study design for identification of OA risk biomarkers at population, precision and personalised risk levels using interpretable AI approaches (SHAP).

the SHAP values, as estimated by the Clin model, for all risk biomarkers across all individuals. The clustering resolution was optimised based on silhouette scores and prediction metrics (Supplementary Fig. 5) identifying 14 clusters of individuals (Fig. 4A). The clustering allowed us to uncover subgroups of individuals predicted to have high risk of OA (cf. prediction probabilities shown in Fig. 4B). Furthermore, by using SHAP values, rather than the original input values, we were able to account for the relative importance of features for OA prediction (Fig. 4C). Finally, all identified clusters were described using the average values of the top 6 features in our model, using the original input values to characterise the differences between clusters. This generated an overview of the most defining characteristics of each OA subgroup, capturing predicted archetypes of OA risk with distinct biomarker profiles (Fig. 4D). Finally, differential expression analyses were performed on the proteomics data (restricted to the subset of OA cases with Olink data, $N = 1723$) to determine which

proteins are differentiating OA cases between each cluster to access molecular
OA-specific risk biomarkers (Fig. 4D, Supplementary Data 3).

To identify clusters with high predicted OA risk and understand subgroup characteristics, we defined the prediction performance of the Clin model within each cluster (F1/positive predictive value (PPV)/ Sensitivity), the percentage of cases in each cluster and average prediction probability (Fig. 5). The top 3 clusters (12, 11 and 0), representing 23% of all individuals, were the clusters for which individuals were best predicted as OA cases, with F1 > 0.83. Another group of 6 clusters (10, 5, 9, 1, 13 and 8; ~35% of all individuals) had more modest values but were still predictive for OA (0.73 > F1 > 0.61). The last 5 clusters (6, 7, 3, 4 and 2; ~41% of all individuals) were the least predictive for OA (F1 < 0.35).

We used a decision tree-based algorithm (SkopeRules[12]) to define sets of rules, per cluster, based on the input values of the Clin model.

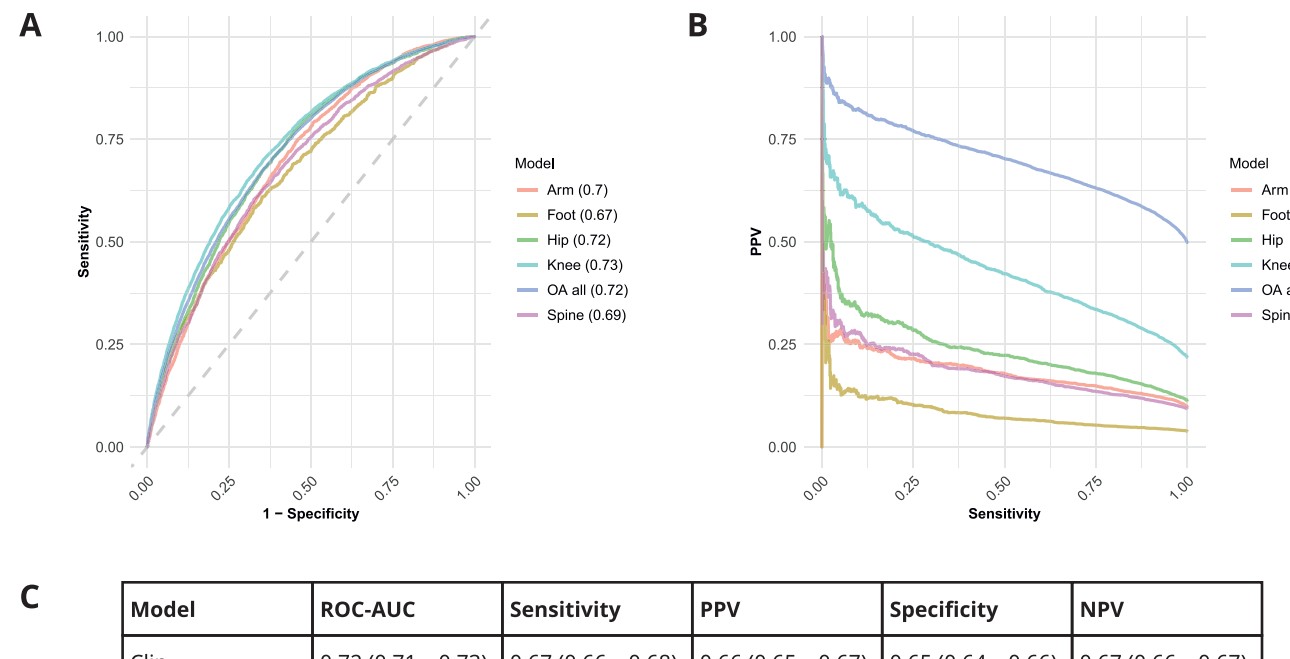

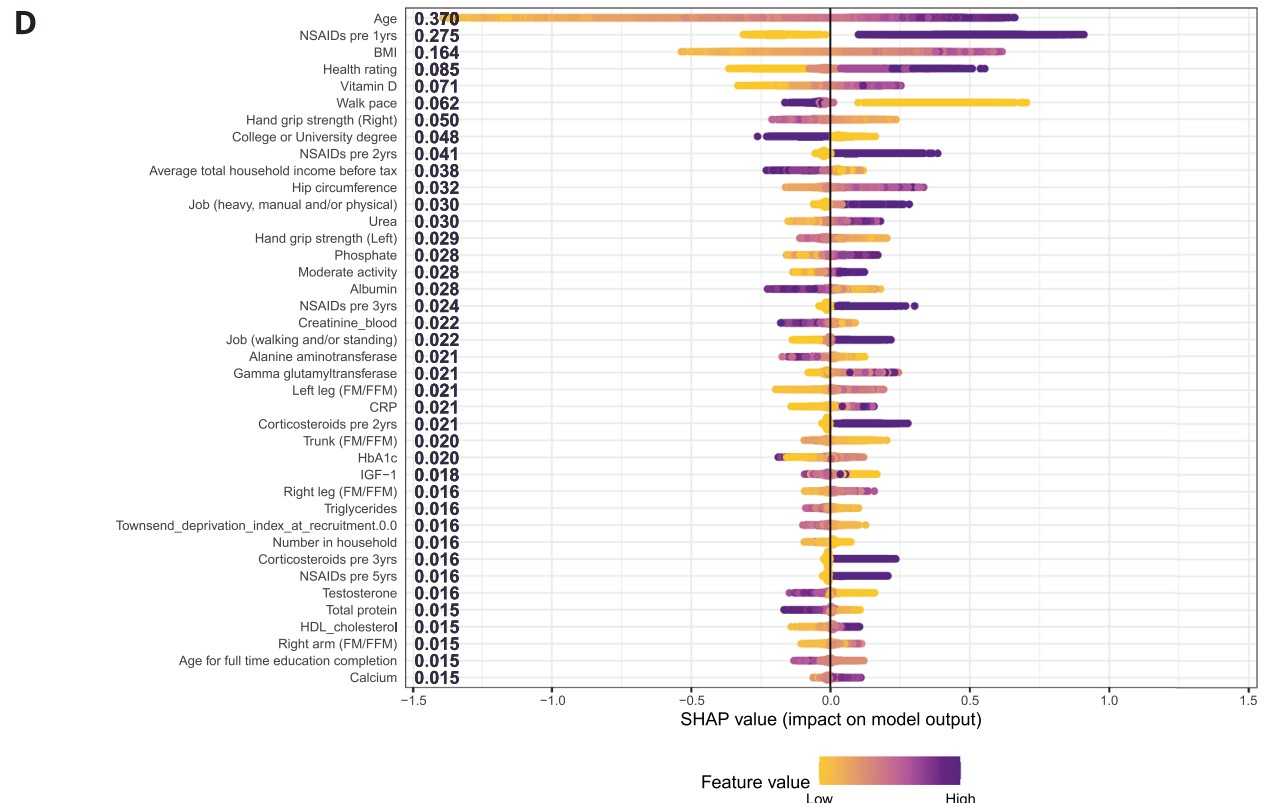

Feature value
Low          High

**Fig. 3 | Clinical prediction model of osteoarthritis (OA) risk (Clin model). A** ROC curves of OA prediction models. **B** Precision-recall curves of OA prediction models. 'OA all' incudes all cases of OA independent of specific joint subsets (Arm, Foot, Hip, Knee, Spine). An OA case can have multiple OA joints affected and a case is included per joint affected (meaning these can be repeated). **C** Performance metrics of Clin model on independent five-fold cross-validation test datasets. PPV positive predictive value, NPV negative predictive value. **D** Ranked feature importance of OA model by SHAP additive explanations for top 40 predictive features in the model. OA osteoarthritis, NSAIDs non-steroidal anti-inflammatory steroid drugs, FM/FFM Fat mass/Fat-free mass.

These rules allowed us to scope each predicted OA archetype by identifying the most distinctive variables and values that determined an individual's cluster allocation with high PPV (Fig. 5, Supplementary Fig. 6). The rules that defined the top 3 high-risk patient groups included: age, prescription of NSAIDs within the last year, hand grip strength, self-reported walking pace and health rating, Townsend deprivation index and IGF-1 levels (Fig. 5). To validate the potential clinical value of these rules, we applied them to an independent hold-

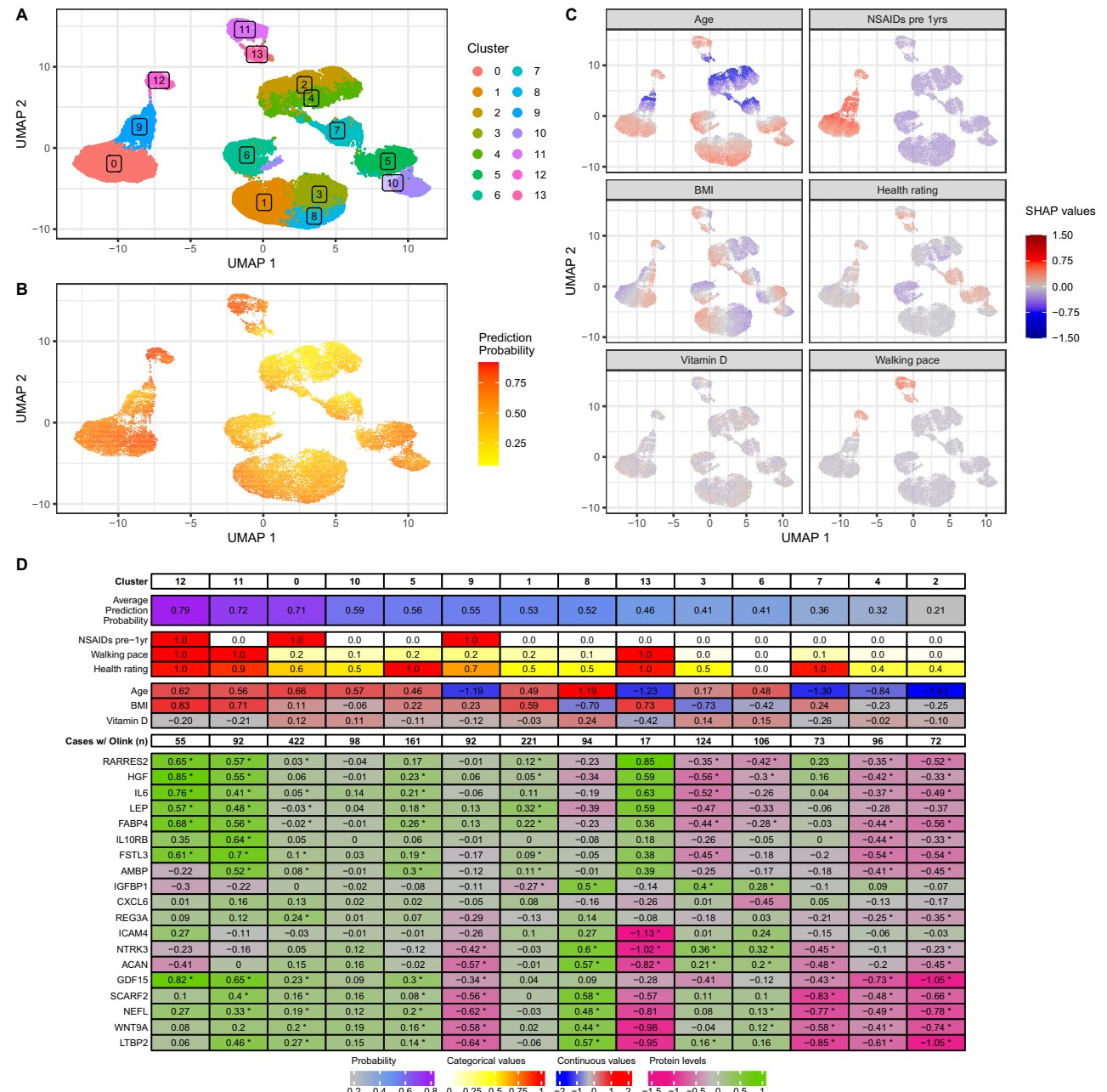

**Fig. 4 | Osteoarthritis (OA) patient clustering and characterisation. A** Clusters obtained based on the SHAP values (Louvain clustering algorithm). **B** OA prediction probability per individuals. **C** SHAP values used for clustering; the colour scheme allows visualisation of the importance of each feature in the prediction model as well as their impact on clustering. For (**B**) and (**C**), points were binned to increase readability; the average values within each of the bins are plotted. Clusters were obtained after dimensionality reduction of the SHAP data with a principal component analysis (10 PCs) and visualised after further dimensionality reduction with UMAP. **D** Average values of prediction probabilities, the top 6 most predictive features in the OA model, and circulating plasma biomarkers levels for the most differentiated proteins between OA cases in each cluster. Categorical values encoding before averaging: NSAIDs (pre-1years): 1 and 0 represent patient taking or not taking the drug, respectively; walking paces were rescaled from 0 to 1 corresponding to a range from fastest to slowest; health ratings were rescaled from 0 to 1 corresponding to a range, from healthiest to least healthy. Continuous values (including proteomics data) encoding before averaging: values were transformed into Z-scores for visualisation purposes. Plasma proteomics biomarkers have been identified using the OA cases in each cluster with available Olink data ($N = 1723$ total) and taking the top 2 most significantly up- or down-regulated proteins per cluster. Proteins can be a biomarker for multiple clusters, resulting in 19 biomarkers for 14 clusters; significant proteins are annotated with an asterisk (*) (adjusted p-value ≤ 0.05, logistic regression adjusted for sex; p-values Bonferroni-corrected per cluster). Full results and exact p-values provided in Supplementary Data 3, as well as cluster sample sizes). OA osteoarthritis, NSAIDs non-steroidal anti-inflammatory steroid drugs.

out validation population (with similar case/control definitions, and with cases being diagnosed more than 5 years after the assessment centre visit (up to 11 years); 7341 cases and 5999 controls). While 4% of individuals could not be accurately mapped to any subgroup based on these sets of rules, we were able to uniquely assign 88.2% of

individuals to a cluster, and 7.8% to two possible clusters. In the latter case, we selected the cluster with the highest percentage of cases in the OA study population in order to minimise the risk of false negatives (Fig. 5 shows cluster attribution). We observed a high correlation in the percentage of cases between clusters of the OA study and

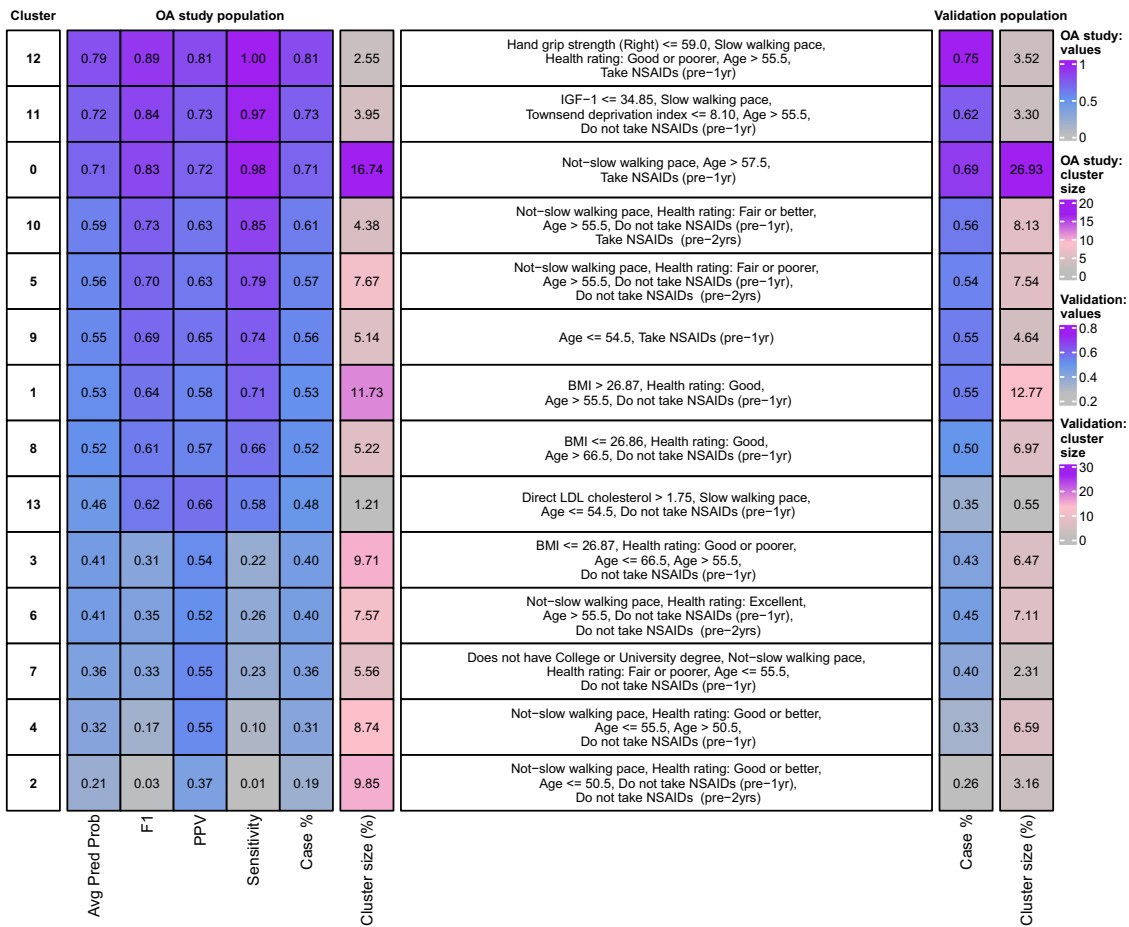

**Fig. 5 | Cluster prediction metrics and defining rules in the osteoarthritis (OA) study population (time-window between data collection at the assessment centre and OA diagnosis (any OA): less than 5 years) and validation in independent hold-out population (time-window: 5 to 11 years).** Left (heatmap): each cluster is defined by prediction metrics, percentage of cases, cluster size (%). Middle (text): set of rules best defining each cluster, based on the model input values and generated by a decision tree model. Right (heatmap): percentages of cases and cluster size (%) in an independent population in which individuals were attributed to clusters according to their corresponding rules. OA osteoarthritis, NSAIDs non-steroidal anti-inflammatory steroid drugs, Avg Pred Prob average prediction probability per cluster, PPV Positive predictive value.

validation populations ($R^2 = 0.90$), as well as a strong correlation in the proportion of individuals in each cluster ($R^2 = 0.68$) (Supplementary Fig. 7).

**Personalised risk biomarkers of OA**

Interpretation using SHAP values from the Clin model enabled quantification of the impact an individual's patient data had on their predicted risk of an OA diagnosis. We extracted and visualised individual OA risk profiles using waterfall plots, demonstrating the predicted positive and negative impact of personal OA risk biomarkers (Fig. 6). For example, a patient who developed OA from Cluster 1 had a predicted risk of 64% for OA. This predicted risk was predominantly driven by a BMI in the obesity range and age of 65 years (Fig. 6A). However, this patient had not taken NSAIDs 1 year prior to OA diagnosis, and this decreased the predicted risk for OA diagnosis. Other risk biomarkers had additional minor contributions to increasing the predicted OA risk, including a lower muscle strength (indicated by hand grip strength), and lower socioeconomic status (indicated by lower average income and level of education). For this patient, if the BMI was not considered, the predicted OA risk would have decreased to 57%. While this approach cannot demonstrate causality between BMI and OA risk, our results are suggestive of potential intervention opportunities on high impact modifiable risk biomarkers that may be driving increased risk prior to an OA diagnosis.

Additional individual OA risk profiles were examined, including an individual from cluster 2 with very low predicted OA risk (Fig. 6B), patient with OA from cluster 12 with multiple signs of poor metabolic health (Fig. 6C), and a younger individual with high BMI and NSAID prescription, with additional lifestyle factors such as heavy manual/physical work and shift work from cluster 9 (Fig. 6D).

**Multi-omics OA risk biomarkers**

To explore molecular risk biomarkers of OA in the context of the clinical prediction model (Clin model), we integrated various types of omics data with the clinical features including OA genetics (ClinSNP, ClinWGPRS, ClinGRS and ClinPath models), metabolomics (ClinMet model) and proteomics data (ClinPro model) for subsets of individuals where this data were available (Fig. 7A; genetic risk scores (GRS) generation described in Methods and Supplementary Fig. 8). The predictive performance remained unchanged compared to the Clin model (ROC-AUC ranging: 0.70–0.72, Fig. 7A). A sensitivity analysis of the Clin model's predictive performance confirmed the performance on these specific omics subsets of patients was also unchanged (Supplementary Table 2). However, the inclusion of OA omics signatures influenced the rankings of OA risk biomarkers in the models (Fig. 7A–E).

Firstly, when including a whole-genome polygenic risk score (WGPRS) for OA in our models (ClinWGPRS), it was the sixth most predictive risk feature for OA risk (Supplementary Fig. 9). At the population level, the OA WGPRS did not affect the overall

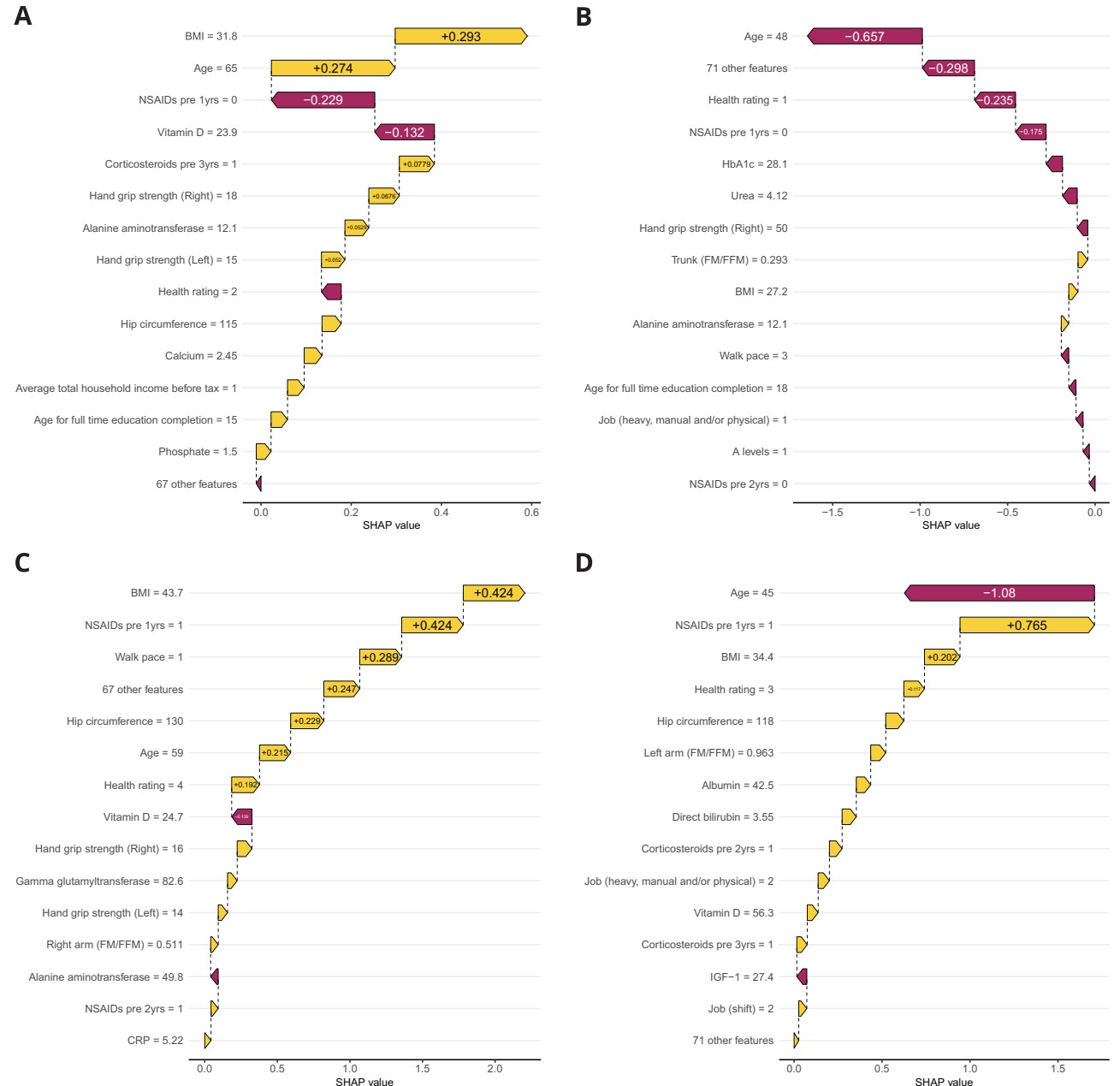

**Fig. 6 | Individual risk profiles.** For example patients from (**A**) cluster 1, (**B**) cluster 2, (**C**) cluster 12 and (**D**) cluster 9. Waterfall plots show the top 15 most important features for estimating the OA risk at the individual level. Yellow bars (positive SHAP value) indicate features that increased predicted OA risk; red bars (negative SHAP value) indicate features that decreased predicted OA risk. Numbers within bars represent the SHAP value for the feature; numbers on the y-axis represent the value for this feature, both are specific to the individual shown and represented the magnitude of the effect of the risk biomarker on predicted OA risk. OA osteoarthritis, NSAIDs non-steroidal anti-inflammatory steroid drugs.

performance of the model (ROC-AUC: 0.72) compared to Clin model. However, the predictive performance for the model (F1 and sensitivity) were higher for a population with higher OA WGPRS (Supplementary Fig. 10). Therefore, although the OA WGPRS provided minimal additional benefit when predicting OA at the population level, it did provide additional value for subgroups with more extreme genetic risk.

At the gene locus level, the highest ranked gene-level GRS by the ClinGRS models were *TGFB1*, *GDF5*, *PTCH1* and *FAM53A* (Fig. 7B, Supplementary Fig. 11). The relevance of *TGFB1* to OA was further supported by the TGF-β signalling pathway being the top ranked pathway-level polygenic risk score (pathway-PRS in ClinPath, Fig. 7C and Supplementary Fig. 12). Other pathways identified as being

predictive of OA, in the context of clinical features, included glycosphingolipid biosynthesis, adipocytokine signalling and cytokine-cytokine receptor interactions. No strong predictive signal was identified for previously reported OA-associated single nucleotide variants[5,13] (ClinSNP). Blood plasma metabolites that were identified as being important for the prediction of OA (ClinMet) included acetate, valine, 3-hydroxybutyrate, citrate and the percentage of saturated fatty acids to total fatty acids (Fig. 7D). Strong predictive proteomic signatures were identified (ClinPro) including CRTAC1, COL9A1, ACTA2, EDA2R and TACSTD2, which were the most predictive of OA risk together with higher age, prescription of NSAIDs during the year prior to OA diagnosis, and higher BMI as seen in the Clin model (Fig. 7E). There were significant differences in the normalised

**A**

| Model name | ClinSNP | ClinGRS | ClinPath | ClinMet | ClinPro |
|---|---|---|---|---|---|
| Omics data | Confident OA SNPs | Gene OA Risk Score | KEGG pathway OA PRS | NMR metabolomics | Olink proteomics |
| Number of omics features | 85 | 193 | 186 | 249 | 1461 |
| Number of Cases/Controls | 18,625 / 18,779 | 18,625 / 18,779 | 18,625 / 18,779 | 4,502 / 4,519 | 1,723 / 1,816 |
| ROC-AUC | 0.72 | 0.72 [1,2] | 0.72 [1,2] | 0.72 | 0.70 |
| Top 1 feature | - | TGFB1 [1,2]* | TGF beta signaling pathway [1,2]* | Acetate | CRTAC1 |
| Top 2 feature | - | GDF5 [1,2]* | Glycosphingolipid biosynthesis globo series [2] | Valine | COL9A1 |
| Top 3 feature | - | PTCH1 [1,2] | Adipocytokine signaling pathway [1]* | 3-Hydroxybutyrate | ACTA2 |
| Top 4 feature | - | FAM53A [1] | Cytokine-cytokine receptor interaction [1]* | Citrate | EDA2R |
| Top 5 feature | - | - | - | Saturated fatty acids to total fatty acids percentage | TACSTD2 |

**Fig. 7 | Multi-omics osteoarthritis (OA) risk models and biomarkers. A** Top ranked features from omics models in the context of multi-modal clinical features. The top five omics features that appeared important for prediction of OA based on the average marginal SHAP value ranking amongst top 40 predictive features. For ClinSNP, ClinGRS and ClinPath, several sensitivity checks were done for the genetic features including results marked with: (1): GRS obtained with proxy, (2): GRS obtained without proxy, and * Identified for models with genetic features corrected for population stratification (Supplementary Table 3 for details). For the column Gene OA Risk Score, italic refers to the defined gene loci. **B** Ranked feature importance of ClinGRS (proxy) model by SHAP additive explanations for top 40 predictive features in the model. **C** Ranked feature importance of ClinPath (proxy) model by SHAP additive explanations for top 40 predictive features in the model. **D** Ranked feature importance of ClinMet model by SHAP additive explanations for top 40 predictive features in the model. **E** Ranked feature importance of ClinPro model by SHAP additive explanations for top 40 predictive features in the model. OA Osteoarthritis, NSAIDs non-steroidal anti-inflammatory steroid drugs, FM/FFM Fat mass/Fat-free mass.

expression levels of all five proteins between OA cases and controls. Higher levels of CRTAC1, COL9A1, ACTA2, EDA2R were observed in patients with OA, whereas lower levels of TACSTD2 were observed in patients with OA. Interaction dependencies in the XGBoost model (ClinPro) between these top five proteins and the most predictive clinical features of OA risk (age, NSAIDs prescriptions 1 year before diagnosis and BMI) were tested. Significant interactions were identified between age and CRTAC1, COL9A1, EDA2R and TACSTD2 (Supplementary Fig. 13). Protein levels of COL9A1 were more important for OA prediction in people with an age above 55. EDA2R was generally seen in

**A**

| Arm | Foot | Spine | Hip | Knee | Rank |
|---|---|---|---|---|---|
| Age | NSAIDs pre 1yrs | NSAIDs pre 1yrs | Age | BMI | 1 |
| NSAIDs pre 1yrs | Age | Age | NSAIDs pre 1yrs | Age | 2 |
| Hand grip strength (Right) | Vitamin D | Health rating | Walk pace | NSAIDs pre 1yrs | 3 |
| Vitamin D | BMI | College or University degree | Hip circumference | Vitamin D | 4 |
| Hand grip strength (Left) | Hand grip strength (Left) | Average total household income before tax | BMI | Job (heavy, manual and/or physical) | 5 |
| Phosphate | Paid employment self employed | Vitamin D | Vitamin D | Average total household income before tax | 6 |
| Left leg (FM/FFM) | Townsend deprivation index | Age for full time education completion | Urea | Walk pace | 7 |
| College or University degree | Albumin | HbA1c | CRP | Moderate activity | 8 |
| HbA1c | Car | Hand grip strength (Right) | Health rating | NSAIDs pre 2yrs | 9 |
| Calcium | Urea | BMI | Triglycerides | Hip circumference | 10 |
| Moderate activity | Potassium | Urea | Total protein | Urea | 11 |
| Job (heavy, manual and/or physical) | Cystatin C | Right arm (FM/FFM) | NSAIDs pre 2yrs | Townsend deprivation index | 12 |
| Townsend deprivation index | Health rating | Albumin | Potassium | A levels | 13 |
| Glucose | Aspartate aminotransferase | Alanine aminotransferase | IGF-1 | CRP | 14 |
| BMI | Total protein | Urate | Phosphate | Health rating | 15 |

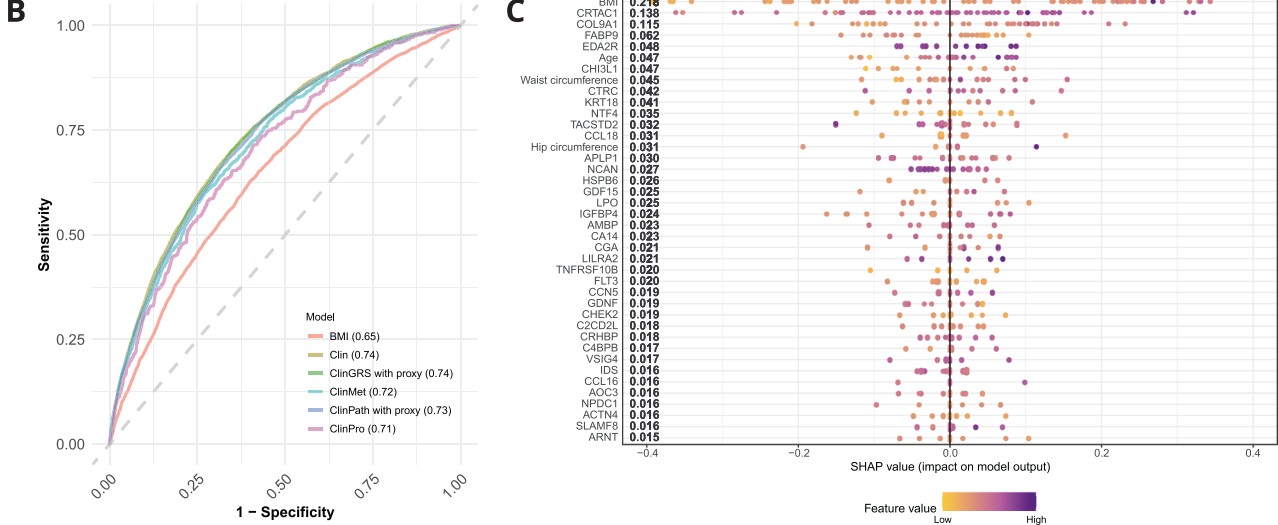

**Fig. 8 | Osteoarthritis (OA) joint-specific models. A** Feature importance of the top 15 features ranked by SHAP additive explanations are provided for joint-specific models of people with diagnosis of OA in the arm, foot, spine, hip or knee. Detailed descriptions of the features listed are found in Supplementary Data 4. **B** ROC-curves of knee-specific models trained only using only patients diagnosed with knee OA using BMI only (BMI), or similar features as in the Clin (5341 patients with knee OA, 19,252 controls), ClinGRS with proxy (5205 patients with knee OA, 18,779 controls), ClinPath with proxy (5205 patients with knee OA, 18,779 controls), ClinMet (1265 patients with knee OA, 4519 controls), or ClinPro (488 patients with knee OA, 1816 controls) models. **C** Ranked feature importance of ClinPro knee-specific model by SHAP additive explanations for top 40 predictive features in the model.

lower protein levels for individuals under 60 years old, while increased importance of EDA2R as an OA risk biomarker was seen in older individuals.

## OA risk biomarker heterogeneity across joints

To further explore impact of risk biomarkers in different joints diagnosed with OA, we retrained our Clin model on subsets of individuals that were diagnosed with OA in any of the five joints identified from the clinical OA diagnoses (Fig. 1B). No major differences in predictive performance were observed when comparing the Clin model to the joint-specific models except for the foot-specific model which had lower predictive performance (ROC-AUC 95%CI for models: arm-specific: 0.70 (0.67–0.72), foot-specific: 0.64 (0.62–0.66), spine-specific:

0.68 (0.67–0.70), hip-specific: 0.72 (0.70–0.74), knee-specific: 0.74 (0.73–0.75)). The models had different rankings of the most predictive features (Fig. 8A and Supplementary Figs. 14–18). Age and prescription of NSAIDs one year ahead of the OA diagnosis was still important for prediction of OA risk stratified per joint. However, BMI had varying importance, with increased importance for predicted risk of an OA diagnosis in weight-bearing joints (knee, hip and foot) compared to arm and spine.

OA patients diagnosed in the knee represented the best-powered OA subgroup and to further explore the importance of BMI relative to multi-omics signals, we retrained the ClinGRS, ClinPath, ClinMet, and ClinPro models for these subsets of patients. BMI on its own predicted OA risk with ROC-AUC 0.65 (95%CI: 0.64–0.66, Fig. 8B). Performance

improved when including additional features beyond BMI, but no major differences in predictive performance was observed across the knee-specific Clin, ClinGRS, ClinPath, ClinMet and ClinPro models (ROC-AUC ranging: 0.71–0.74, Fig. 8B). The knee ClinPro model resulted in ROC-AUC: 0.71 (95%CI: 0.68–0.75) with BMI being ranked the most important feature for prediction of knee OA risk. CRTAC1, COL9A1 and EDA2R were still observed amongst the top 10 predictors of OA risk, as was seen in the ClinPro model (Fig. 8C). Other proteins also important for prediction in the knee-specific ClinPro model included FABP9 and CHI3L1, CTRC and KRT18.

## Discussion

We present a large-scale study of OA, in a UK cohort of ~20,000 patients with OA and ~20,000 controls, encompassing a broad set of longitudinal OA risk biomarkers. We utilised interpretable machine learning to address gaps in the field of prediction and understanding of OA pathogenesis. Our models advance our understanding of risk of OA diagnosis, generating hypotheses for early screening, prevention and treatment of OA. The presented models predict an individual's 5-year risk of an OA diagnosis from EHR data which encompasses the range of OA heterogeneity and pathophysiology in real-life clinical settings. The complexity of OA risk was captured using multi-modal clinical data, biochemical and molecular signatures of OA. We identified distinct subgroups of OA risk profiles and derived simple clinical association rules for these subgroups. We also mapped differentially expressed molecular biomarkers between OA cases in risk subgroups. Finally, we demonstrated how individual patient journeys can be dissected to identify risk biomarkers for OA, contributing to the development of personalised preventative strategies.

We present a retrospective case-control study that extracted 5 years of longitudinal data prior to OA diagnosis, or control index date, for risk modelling. A strength of this retrospective design is the use of all available data 5 years prior to OA development. This contributes to our understanding of potential windows for preventative interventions, identifying risk biomarkers when patients are at high-risk. The Clin model demonstrated strong performance, aligned with previously published models[14]. Previous models that reached above ROC-AUC 0.75 usually included imaging variables (e.g., X-rays), pain scores, or data regarding osteoporosis and previous leg injury[15–17]. These variables were not available in our model and might impact real-life applicability, as this data may not be widely available in primary care. The comparison to other published OA models is limited by differences in prediction horizons, age and sex distributions, and OA definitions. Our model considered a diverse OA phenotype identified from EHR codes across patients diagnosed with OA, across mostly undifferentiated joints specifications. Furthermore, it quantified the predictive impact for a range of OA risk biomarkers, including genetics, clinical biomarkers and environmental factors, to which previous studies have provided limited insights[4,8,9]. Some of these risk biomarkers are recognised as OA risk biomarkers (e.g., age and BMI)[11], while others are not traditionally considered OA risk biomarkers (e.g., personal health rating, hand grip strength, body composition, and walking pace). This highlights the novelty of the OA risk models in our study, offering new opportunities for prevention by addressing novel modifiable risk factors. These biomarkers improved the predictive performance and confidence across individual-level predictions, compared to a simpler model based only on well-known OA risk biomarkers (age, sex and BMI). Preventative interventions may need to target multiple risk biomarkers to reduce OA incidence. Most of the predictors included in the Clin model are easy to obtain in clinical practice, although some may require specific testing (e.g., hand grip strength and regional body composition) which might be challenging depending on the health care system and the setting. Risk biomarkers that are easy to obtain clinically may enable preventative strategies that prioritise interventions to those with the highest risk.

BMI is an established risk biomarker for OA[18] and was the 3rd predictor of OA in the Clin model. The impact of BMI on the risk of OA was present even in those who did not have BMI ≥ 30 kg/m$^2$, with lower BMI levels reducing OA risk. This is unsurprising since increases in BMI levels, regardless of the BMI category in which the increase occurs, can increase the risk of obesity-related complications, as seen in type 2 diabetes[19]. This is possibly due to the presence of a personal fat threshold, as it has been seen that weight loss can lead to type 2 diabetes remission even in those who do not have obesity[20].

NSAID prescription 1 year prior to OA diagnosis was the second most important predictor. This likely reflects the delay or clinical inertia in the diagnosis of OA, reflecting several barriers to OA management in primary care that have been described previously[21,22]. The model was trained to predict OA diagnosis, as indicated by EHRs. To minimise the risk of contamination of OA patients in the control group we used previously published clinical codes[23], in addition to a curated search for OA codes, when defining the OA study and validation populations. However, it is likely that more severe OA cases are captured with a clinical diagnosis. Hence, control contamination is most likely due to less severe OA cases, potentially minimising the impact on model performance and outputs.

Additionally, lower socioeconomic status, indicated by a lower income and level of education, was predictive of an increased OA risk. This agrees with previous work, that demonstrated that social deprivation, including lower education, was associated with increased OA risk[24,25]. Lower social deprivation and increased OA risk were previously linked to obesity[24].

Sex is an established risk factor for OA, with women being at higher risk of developing OA compared to men[1]. In our Clin model, sex only modestly contributed to prediction of OA, being ranked the 85th most predictive feature. This is likely because the Clin model included other features that potentially reflect sex as an OA risk factor. Sex was strongly correlated (R$^2$ > 0.75) with testosterone (ranked 35th most predictive feature) as well as all measurements of body composition metrics of fat mass to fat free mass (ranked between 24–60th most predictive features). Based on SHAP, lower total testosterone and higher fat mass/fat-free mass ratio were associated with higher risk of OA and female sex. It is also possible that sex was not highly ranked due to the complex relationship with vitamin D levels. In this study, higher vitamin D levels were associated with higher risk of OA. Our analysis showed that individuals that reported taking vitamin D supplements were older and had higher levels of vitamin D. Additionally, a higher proportion of individuals taking vitamin D supplements was observed amongst post-menopausal women (a known high-risk group for OA[26,27]), compared to men or pre-menopausal women. Therefore, vitamin D levels in the Clin model may be capturing an older population and post-menopausal women who take vitamin supplements. These findings suggest that other parameters in the model account for not identifying sex as a highly ranked in our model.

Disentangling these risk biomarkers at an individual level has previously been limited by differential impacts of various risk biomarkers at the individual level. A strength of our study is using interpretable AI approaches (SHAP estimation) for quantifying the impact of individual risk biomarkers, in the context of all integrated variables, at a subgroup and individual level. This might aid the prioritisation of future preventative strategies, by selecting the modifiable factors with highest contribution to risk. However, this needs to be tested in clinical trials. Our study cannot assess whether intervention on risk biomarkers would decrease OA risk, the extent of this decrease, or the causal relationship between the risk biomarkers and OA. However, it has previously been shown that reduction in modifiable risk biomarkers corresponded with a reduced individual probability of developing knee OA[10].

In our study, most participants had an undifferentiated diagnosis of OA, potentially introducing noise in the model, reflecting

heterogeneous biological mechanisms and phenotypes[5,6]. To address this further, we explored joint-specific models and concluded, in line with previous studies, that BMI is a higher risk biomarker for OA in weight-bearing joints, among other joint-specific risk biomarker profiles. However, future studies in larger subset of patients with joint-specific effects should further validate these findings and allow better understanding of joint-specific OA pathogenesis.

There have been efforts to identify subgroups of OA to enable targeted treatment and management of patients. Previous studies are limited by using restricted sets of biomarkers, narrowly defined OA populations, and by identifying subgroups of established/diagnosed disease[6,28]. By modelling OA risk biomarkers prior to diagnosis, we identified opportunities for early intervention and prevention. Our results suggest that simple sets of rules may be used to assign most individuals to specific subgroups, which differ in their predicted OA risk and risk biomarkers. This may inform clinicians and patients in assessing the potential for belonging to high-risk groups, the threshold for initiating an OA diagnostic assessment, and identifying the most important risk biomarkers to address. These subgroups were validated in a hold-out population evaluating the 11-year OA risk (5- to 11-year window). A high consistency in the subgroups was observed across the 5-year and 11-year risk populations, indicating that the major risk biomarkers, and associated risk profiles, remained relevant for the prediction of OA diagnosis across these time periods. Therefore, interventions addressing the modifiable risk factors may have an impact on OA risk, although this requires further testing.

Although the integration of omics did not improve overall model performance, it did change risk biomarker ranking. Other studies have included genetic factors, but also failed to improve model performance[15,29]. A strength of our study is the interpretable AI, with the impact of omics highlighting relevant biological pathways and guiding OA-specific prevention strategies. Currently, omics data may not be readily available in routine clinical practice; however, this may change as the utility of omics biomarkers matures.

In our models, some omics biomarkers replaced clinical risk biomarkers, with some of them being more predictive of OA risk than BMI. CRTAC1 was the most predictive protein and has previously been proposed as an OA risk biomarker[30-32]. CRTAC1 is associated with OA diagnosis across multiple joints and with the severity of OA[30-32]. It has been suggested that CRTAC1 is upregulated in joints in OA by pro-inflammatory cytokines[33]. COL9A1 was also predictive of OA diagnosis and is both genetically and epigenetically linked to OA[33-37]. Additionally, mutations in COL9A1 are associated with multiple epiphyseal dysplasia, a hereditary condition characterised by early onset OA[33]. Another predictive protein was ACTA2[10], which has previously been associated with subgroups of OA[38] and is associated with clusters of smooth muscle cells in the OA synovium[39]. Lastly, the predictive protein EDA2R has previously been associated with TNF mediated inflammation, in the context of rheumatoid arthritis[40].

In the knee-specific ClinPro model, other proteins predictive of OA included FABP9, CHI3L1, KRT18, and CTRC. The function of FABP9 is insufficiently understood, with no direct link with OA[41]. However, other proteins in the FABP family are involved in inflammatory response and oxidative stress[42,43]. CHI3L1 is involved in tissue injury, inflammation, tissue repair, and is associated with OA[44]. KRT18 has no known association with OA but has been suggested as a biomarker of intervertebral disc degeneration in vitro[45]. Lastly, CTRC is a serum calcium-decreasing factor and may be involved in LPS-induced inflammatory responses[46].

We also explored which proteins were differentially expressed across subgroups of OA risk. These may reflect differences in relevant biological pathways such as inflammation (e.g., IL6, CXCL6), body weight regulation/energy homoeostasis (e.g., LEP, GDF15), but also in proteins directly relevant to osteoarthritis, such as the proteoglycan

ACAN (aggrecan)[47] and AMBP (bikunin precursor)[48]. Furthermore, HGF, which has previously been shown to be involved in both obesity[49] and OA biologies[50], is significantly over-expressed in OA cases from most clusters with higher BMI and under-expressed in most lower BMI clusters. Therefore, despite HGF being associated with OA in osteoblasts, its plasma levels may be more heterogenous.

Multiple genetic risk variants are associated with OA[5]. Although the OA WGPRS had limited predictive value across the general population, we found it may be informative for those with more extreme genetic risk. Only a small proportion of variance in OA can be explained by genetics[5] and, in prior work, genetics was limited in predicting OA[15,51]. This highlights the importance of including clinical and biological data in prediction models, to provide contextual disease information.

Our ClinGRS model enabled us to estimate the contribution from OA-associated genes on OA risk. The *TGFB1* locus was a predictive feature, and TGF-β signalling affects multiple cell types in OA development and progression[52]. Multiple components of the TGF-β pathway have been genetically associated with OA[52,53]. In phase 3 clinical trials, TGF-β1 cell and gene therapy improved function and pain in knee OA[54]. The *GDF5* GRS was predictive for OA and GDF5 has a role in chondrogenesis[55]. Higher levels of GDF5 were seen in OA with advanced cartilage damage[56], and GDF5 is currently a target in clinical development for cartilage regeneration indications[53]. Lastly, the *PTCH1* GRS was predictive of OA, and *PTCH1* encodes a receptor for Hedgehog ligands. Hedgehog signalling is associated with multiple OA mechanisms and disease severity in OA[5,57,58]. *PTCH1* is genetically associated with total hip, knee and joint replacements[5], and targeting the Hedgehog pathway may be a therapeutic opportunity for OA[57].

Multiple metabolites were relevant for predicting OA diagnosis and have previous associations with OA metabolism. Acetate was the most predictive metabolite for OA. ACOT12 breaks down acetyl-CoA into acetate and CoA, and ACOT12 is a novel regulator of de novo lipogenesis (DNL) associated cartilage degradation in OA[59]. Valine was also predictive for OA and is an essential branched chain amino acid (BCAA). Valine has previously been associated with the severity of inflammation in synovial tissue[60]. BCAAs may play a role in increased inflammation, reduced autophagy, and increased insulin resistance in OA[61]. The ketone body 3-hydroxybutyrate (βHB, β-hydroxybutyrate) may have anti-senescence effects which delay OA progression[62]. Finally, citrate levels in synovial fluid have conflicting associations with knee OA[63] and may be linked to altered energy metabolism[64].

The relationship between fatty acids and OA may be impacted by sex, obesity, OA joint and whether measured in the fasted or post-prandial state[65,66]. Different fatty acids have distinct effects on OA[67]. Longer chain saturated fatty acids induce metabolic syndrome and OA-like cartilage degradation[68,69] and systemic inflammation, independent of weight-gain, in obesity related OA[70].

In summary, we present a retrospective study using large-scale predictive models for the diagnosis of OA, incorporating a broad range of risk biomarkers. The use of interpretable machine learning for individual patients enabled the identification of personalised modifiable risk biomarkers, opening the opportunity for tailored OA preventative strategies. Our integration of omics features identified OA-specific risk biomarkers and highlighted the predictive importance of underlying OA disease biology. Taken together, these findings may advance early screening, prevention, and treatment of OA, reducing both disease incidence and progression. External validation of the identified risk biomarkers and models in an independent cohort is needed to investigate the replicability of the model and identified subgroups. Finally, validation in a range of cohorts representing a diversity of genetic, cultural backgrounds and healthcare practices would further our understanding of the impact of this contextual information on OA risk.

## Methods

### UK Biobank

The UK Biobank is a human cohort comprising ~500,000 individuals, aged between 40 and 69 years at recruitment, from 2006 to 2010[11]. At a recruitment assessment centre, participants contributed to in-depth data collection and banking of biological samples. Biological samples were subsequently used for the generation of various omics data and biomarker measurements. Clinical outcomes for participants can be followed-up through linkage to secondary health care data (hospitalisations, available data until 03/2017) and for ~45% of participants primary health care data (general practices, available data until 09/2017). The machine learning models integrated both clinical, metabolomics, proteomics and genetic data from the recruitment assessment centre as well as longitudinal information captured from the EHR data.

### Ethical approval

UK Biobank is available to researchers following application to the UK Biobank database (https://www.ukbiobank.ac.uk/enable-your-research/apply-for-access). The UK Biobank has ethical approval from the North West Multi-centre Research Ethics Committee (REC reference number: 16/NW/0274). Written informed consent was obtained for all UK biobank study participants. Analyses for this study were conducted under application numbers 53639 and 65851.

### Pre-processing of assessment centre data

**Clinical assessment centre data.** Diverse participant data from the recruitment assessment centre was processed for input into machine learning models to integrate multi-modal signals in the models (Supplementary Data 4). This included data on socio-demographics (including Townsend deprivation index), lifestyle, diet, and physical activity. Additionally, a panel of blood and urine biomarkers were measured at recruitment (Supplementary Data 4) and included as input features. Biomarker data from recruitment was processed to assign outliers outside the 1st and 99th percentiles to the value of the 1st and 99th percentile. Body impedance data reflecting body composition was taken from the first assessment instance and included derived measurement of the fat mass (Kg) to fat-free mass (Kg) ratio across whole body, and specific body areas including trunk area, legs, and arms (legs and arms indicated by differences in right and left side of an individual's body composition).

**Genetic data.** The imputed genetics data was generated and provided by the UK Biobank and processed using *PLINK (v.2.00a3LM)*[71,72]. Single nucleotide polymorphisms (SNPs) were subsequently removed if they had a minor allele frequency (MAF) < 1%, a missingness >1%, an imputation score (INFO) < 0.8 or with a Hardy-Weinberg Equilibrium exact $p < 0.00001$. This resulted in a set of around 9.4 M high quality imputed common genetic variants that were used to generate genetic risk scores (GRS).

Genetic information was integrated into the machine learning models through several representations of the genetic information by (i) individual SNPs, (ii) whole genome polygenic risk score of OA (WGPRS) as well as genetic risk scores for either (iii) specific gene-level genetic risk scores (gene-GRS) or (iv) pathway-level polygenic risk scores (pathway-PRS). An overview of the generation of the genetic risk scores is presented in Supplementary Fig. 8

Genetic risk scores were generated using the software *PRSice (v.2.3.3)*[73] and the weights of variants associated with OA from a recent meta-analysis[5] (summary statistics of a sub-analysis excluding the UK Biobank samples).

For gene-GRS, SNPs within 5 kb upstream and 1.5 kb downstream of a gene were used for scoring after clumping. Furthermore, analyses were run with and without a linkage disequilibrium proxy (LD proxy): including SNPs in LD with the clumped region, with $R^2 > 0.8$. The gene

regions were defined using GENCODE annotations (v.43lift37); only protein-coding genes were considered for analyses.

For pathway-PRS, SNPs within genes (as described above) belonging to specific gene-sets were aggregated together into a score (with and without LD proxy). The gene-sets used for the pathway-PRS were generated using KEGG pathways obtained via the Molecular Signature Database (MsigDB; v.2022.1).

The default *PRSice* clumping parameters were used. For WGPRS and gene-GRS/pathway-PRS, no *p*-value based thresholding was applied to maximise the number of SNPs included in the analyses.

SNPs and genes to be included in the models were guided by using previously identified variants and loci associated with OA risk genes from two recent large GWAS-based studies[5,13]. Indels were removed from individual genetic variants, resulting in a set of 85 SNPs. Specific gene-GRS features to include were prioritised based on the genes annotated to the GWAS-significant loci. This included 77 high effector OA genes identified by ref. 5 and 134 OA genes annotated to genome-wide significant SNPs by ref. 13. In total, 204 unique genes associated to OA risk were identified, where 193 genes were protein-encoding and prioritised for the list of gene-GRS used. All 186 pathways from KEGG were tested for pathway-PRS.

**Metabolomics data.** Circulating metabolite biomarkers from blood EDTA plasma samples were quantified from a subset of UK Biobank participants ($N = 118,021$) using Nightingale's high-throughput proton NMR metabolomics platform[74–76]. A total of 249 metabolites were quantified (168 metabolites given in absolute molar concentration units (differs per biomarker, see details: https://research.nightingalehealth.com) and 81 ratios of these) spanning amino acids, (apo-)lipoproteins (incl. subclasses), cholesterol, cholesteryl esters, fatty acids, biomarkers of fluid balance, glycolysis related metabolites, inflammation biomarkers, ketone bodies, phosphor- and/or other lipids. Metabolites that were measured by absolute concentrations were normalised by square root. Metabolites given as ratio or percentages were not processed further before including in the machine learning models.

**Proteomics data.** A subset of UK Biobank participants (as previously described[77]) had biomarkers measured from plasma blood samples by Olink Proximity Extension assays across 58,634 samples for 54,308 individuals. The data was filtered to only include individuals with samples obtained from the first assessment centre, samples were processed and no withdrawn consent as per reported by UK Biobank (48,040 samples and 46,673 individuals). In total, 1472 proteomics markers were measured across four Olink panels (cardiometabolic panel: 369, inflammation: 368, neurology: 367, oncology: 368). The quality of the assay and sample quality was checked by Olink's built-in quality control. Sample with <85% of assays not completely passing Olink's built-in quality control were excluded for analysis. The protein expression was normalised into Normalised Protein eXpression (NPX), which is Olink's arbitrary unit (Log2 scale). NPX values were compared across individuals and extreme outliers were removed if the mean NPX for an individual was <0.2% or >99.8% percentile of the overall cohort mean NPX values. Proteins that were identified by non-normal/bimodal residuals from age, sex, BMI were removed from the analysis. CXCL8, IL6, and TNF were measured across all four Olink assays and an average NPX value was calculated. Furthermore, some samples were assayed twice for replication and for machine learning analyses, an average NPX value was calculated.

### Pre-processing of electronic health record data

Longitudinal data was captured 5 years prior to OA diagnosis/matched index date from the primary and secondary health care data to capture information of biomarkers and medication during the 5 years pre-index date (Fig. 1A and Supplementary Figs. 1–2). Primary care data was

used as a source of longitudinal biomarkers, clinical measurements, and prescription data. Biomarkers and clinical measurements included haematological measurements, liver biomarkers, anthropometric measurements, cardiovascular biomarkers, renal biomarkers, bone & joint biomarkers, lifestyle measurements, hormonal measurements, and diabetes biomarkers. Two EHR coding systems are used in UK primary care (GP) data: Read v2 and CTV3/Read v3 codes (CTV3/Read v3 represent an updated coding system). Both are used for filtering of clinical data in this study, see Supplementary Data 5 for list of codes used. Where appropriate, data was processed to align units and outliers. Medications for type 2 diabetes, obesity and OA was extracted from prescription data (BNF and dm+d clinical codes in Supplementary Data 6).

Longitudinal data was captured in yearly snapshots 5 years prior to OA diagnosis. For continuous data, this was represented as the median value across an 11-month period in the year. The 11-month period was used to capture the majority of the available data from the year, while also leaving a time-gap to avoid overlap between snapshots and the diagnosis/index date. For medication data, prescriptions were grouped into medication classes and a binary value indicated whether the patient had a prescription for the medication class during the year.

Finally, primary and secondary health care data was used to indicate if OA patients had a diagnosis of posttraumatic OA before onset of their OA diagnosis, as this has been shown to increase individual OA risk[78]. Diagnoses of post-traumatic OA were identified and annotated if any codes were given ahead of the OA date/control proxy day.

## OA study and validation populations

OA diagnoses were identified from primary and secondary health care data linked to individual participants (>18 years old) using Read v2, CTV3/Read v3, ICD-9 and ICD-10 clinical codes (Supplementary Data 1). This list was generated using a published list of OA read codes[23], a curated search for OA EHR codes in UK Biobank and review by authors with extensive clinical experience in the UK. Further inclusion criteria for the study included filtering of patients with available primary care data in the UK Biobank. An equal number of control participants, who never developed OA and had observational data during the entire study period (e.g., due to death), were identified. Controls were date-matched with the OA diagnosis dates for case patients to enable retrospective capture of longitudinal data 5 year before OA diagnosis or OA proxy date. Cases and controls were then filtered for those with an OA diagnosis/matched index date after the date of the recruitment assessment centre, and maximum 5 years between the OA diagnosis and assessment centre, allowing the use of this recruitment data in predictive models to estimate the five-year risk (Fig. 1A).

## Machine learning models

Extreme Gradient Boosting (XGBoost) models with decision-tree based learners were implemented by *R v.4.2.0* using the publicly available R libraries; *xgboost (v.1.6.0.1)*, *pROC (v.1.18)* and *caret (v.6.0-93)*. XGboost models were trained using a nested two-level five-fold case-control stratified cross-validation. Missing values were handled by imputation within each cross-validation fold based on the median value available across cases and controls in the training dataset. Features with >50% missing data were excluded from analysis. Additionally, near zero variance features were excluded using near0var from the *caret* library. Hyperparameters for the cross-validated models were optimised via a cross-validation grid search, where optimal parameters were selected based on highest cross-validated ROC-AUC in the inner cross-validation fold. The grid optimised for the following parameters *eta*: (0.05, 0.10, 0.15, 0.20, 0.25, 0.30) and *nrounds*: (50, 100, 200, 300, 500, 700, 1000) with the following other parameters fixed in the XGBoost model; *booster: gbtree, max depth: 10, eval_metric: logloss, objective: binary:logistic, subsample = 0.8, sample:method = uniform* and *min_child_weight = 50*. For models where omics data (genomics,

metabolomics, or proteomics) were integrated the XGBoost model also had *colsample_bytree = 0.8*. The short depth of each decision tree was set to help guide feature selection and reduce risk of overfitting. The selected hyperparameters were selected for re-training of one model in the outer cross-validation level.

The model performance was validated on the outer test set that has been held out from any model training. Model performance was estimated by evaluating classification of true positive (TP), false positive (FP), true negative (TN) and false negative (FN) classifications done by the machine learning model at a standard threshold of 0.5. To account for variance between the five cross-validation sets, both the average and 95% confidence interval (CI) across the five validation datasets were reported. The classification performance is reported by the area under the receiver operating characteristic curve (ROC-AUC), sensitivity (recall, ($\frac{TP}{TP+FN}$)), positive predictive value (PPV, precision, ($\frac{TP}{TP+FP}$)), specificity ($\frac{TN}{TN+FP}$) and negative predictive value (NPV, $\frac{TN}{TN+FP}$). ROC and precision-recall curves were generated using the R library *yardstick (v.1.1.0)*. The robustness and stability of the machine learning models was evaluated across 100 random model initialisation and against 100 random model initialisations with a permuted prediction outcome of OA cases and controls that both accessed impact on predictive performance. Transparent reporting of a multivariate prediction model for individual prognosis or diagnosis (TRIPOD) checklist for prediction model development were followed to ensure clarity and reproducibility of the multivariate prediction model of individual OA risk (Supplementary Data 7).

## Model interpretation

To explore how each individual features contribute to the risk prediction of OA, Shapley Additive explanation (SHAP) values were calculated from the XGBoost models using Tree SHAP[79]. The SHAP values were given as the log-odds of the individual contributions and were visualised using the R libraries *SHAPforxgboost (v.0.1.1)* and *shapviz (v.0.4.1)*. For global estimation of feature importance across the five outer test datasets, the mean absolute SHAP values were calculated, which quantifies, on average, the magnitude of a feature's ability to predict individual risk of osteoarthritis.

## Cluster analyses

To explore differences in feature importance in the prediction model between subgroups of individuals, clustering was performed on the SHAP values. For this, the *Seurat*[80] *(v.4.3.0)* implementation of the Louvain clustering algorithm[81] was used after reducing the data to 10 dimensions by principal component analysis (PCA). A three-steps approach was used to determine the number of clusters. First, the package *chooseR (v.12062020)*[82] was used to perform a subsampling-based approach to generate silhouette scores (as a measure of cluster robustness) for various resolutions (number of clusters). Secondly, the PPV, sensitivity and F1 (a balanced metric evaluating accuracy of case classification, $\frac{2}{sensitivity^{-1} + PPV^{-1}}$) values of each cluster were calculated, based on the prediction results of the XGBoost model. Finally, these two steps were integrated by looking at the cluster resolutions leading to the best combination of these metrics as per Eq. 1:

$$cluster\ score = \frac{weighted\ mean\ (silhouette\ scores)}{1 - weighted\ median\ (prediction\ values)} \quad (1)$$

After manually selecting the optimal resolution parameter to maximise cluster robustness, number of clusters and per-cluster F1 values (resolution = 0.5, $n = 14$ clusters), each cluster was first characterised by averaging the values of the top 6 predictive features to plot as a heatmap using the *ComplexHeatmap* R package *(v.2.13.1)*[83]. Secondly, the *SkopeRules* Python package *(v.1.0.1)*[12] was employed to generate interpretable rules to define each cluster using the original input values used in the XGBoost model (*n_estimators: int = 10, recall_min:*

*float = 0.2, max_depth: int = 5, max_depth_duplication: int = 7*). When multiple set of rules were given for a cluster, the rules appearing in the highest numbers of decision trees were chosen, and the rules with highest out-of-bag PPV and sensitivity in case of equality. Differential expression analyses were performed on the proteomics data (Olink) to identify the most differentially expressed protein per cluster. For this, *Seurat's FindAllMarkers()* function was employed (*min.pct = 0, only.pos = FALSE, test.use = "LR"*). The analyses were corrected for sex as a covariate; *p*-values were adjusted for multiple comparison using Bonferroni correction, per cluster. The full list of differentially expressed proteins per cluster is available in Supplementary Data 3. Analyses were performed in subsets of the samples: OA cases-only (Fig. 3, Supplementary Data 3), controls-only and all samples (Supplementary Data 3).

### Reporting summary
Further information on research design is available in the Nature Portfolio Reporting Summary linked to this article.

## Data availability
UK Biobank is available to researchers following application to the UK Biobank database (https://www.ukbiobank.ac.uk/enable-your-research/apply-for-access). All field IDs and clinical codes used for extraction of data have been provided in Supplementary Data. The use of UK Biobank for this study was performed under research application numbers 53639 and 65851. Source data that is not patient-sensitive data are provided with this paper to reproduce figures and tables in the main manuscript in https://github.com/novonordisk-research/xOAML, including links to any publicly-available datasets used in the analyses. In the case of individual-level sensitive data from UK Biobank mock data describing input file formats is provided alongside code used to reproduce the figures and tables. The OA GWAS summary statistics files used to generate the genetic risk scores were created as part of a published study by Boer et al., Cell 2021.

## Code availability
All analyses were performed on publicly available software, and all parameters are provided in methods wherever relevant. The code used for this study was tailored to the UK Biobank data and is no use as a standalone without access to the UK Biobank. However, code to reproduce the machine learning model, figures as well as files describing input data formats, are provided at https://github.com/novonordisk-research/xOAML. The authors welcome being contacted to provide more information to reproduce the results presented in this paper if needed.

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

## Acknowledgements

The work was supported by research grants from the Innovation Foundation Denmark (RLN) and the Danish Diabetes Academy, which is funded by the Novo Nordisk Foundation, grant number NNF17SA0031406 (RLN). We also thank Eleftheria Zeggini (Helmholtz Munich/Wellcome Sanger Institute) for providing summary level results from the genetic analyses published in ref. 5 for the analysis subset excluding the UK Biobank samples. We also thank the UK Biobank participants and researchers for providing this research resource.

## Author contributions

Substantial contributions to the conception or design of the work: R.L.N., T.M., M.W., A.A.T., Z.M., and R.G. Acquisition, analysis, or interpretation of data: R.L.N., T.M., R.R.K., L.E., L.G.L., A.T.H.S., F.S.G., C.S., M.H., L.S., M.W., A.A.T., Z.M., and R.G. Creation of new software used in the work: R.L.N. and T.M. Drafted the manuscript: R.L.N., T.M., and Z.M. Substantively revision of the submitted manuscript: R.L.N., T.M., R.R.K., L.E., L.G.L., A.T.H.S., F.S.G., C.S., M.H., L.S., M.W., A.A.T., Z.M., and R.G. Approval of the submitted manuscript: R.L.N., T.M., R.R.K., L.E., L.G.L., A.T.H.S., F.S.G., C.S., M.H., L.S., M.W., A.A.T., Z.M., and R.G.

## Competing interests

R.L.N., T.M., R.R.K., L.G.L., A.T.H.S., F.S.G., C.S., L.S., M.W., A.A.T., Z.M., and R.G. are employed by Novo Nordisk and own minor company stock. L.E. is currently employed by Nordic Bioscience A/S and was working on this manuscript while being employed by Novo Nordisk A/S. M.H. declares no competing interests.
