## [Peer Review File · Nature Communications]

Data-driven identification of predictive risk biomarkers for subgroups of osteoarthritis using interpretable machine learningREVIEWER COMMENTS

Reviewer #1 (Remarks to the Author):

Summary:

Osteoarthritis is a common, debilitating disease with no approved treatments for preventing disease onset or slowing progression. Most patients are identified in end-stages where the only option is joint replacement surgery. It is well known that OA is heterogeneous disease and using machine learning approach to refine the disease into subtypes may provide clarity needed to develop personalized therapies. To address this highly relevant and important problem, the authors for the current manuscript "developed a machine learning model to predict individual five-year risk of OA and identify risk biomarkers". A retrospective case control design was harnessed to identify OA cases and controls with appropriate longitudinal data. They also performed an interesting subgroup analysis identifying 14 sub groups which were validated in an independent set of patients. I found this to be a very interesting analysis overall but do have some significant comments and questions. A major missed opportunity was that the multi-omics data was not used to disentangle the role of BMI in OA. High BMI is an established predictor of OA but the underlying reasons BMI influences OA etiology and progression are likely diverse. Some patients may benefit from classic weight reduction and others from medication to reduce inflammation or even some other unknown pathways the field has not yet attributed to BMI role in OA. Further, the findings from multi-omics and clinical prediction models overall were not integrated in such a way to advance the field. Overall, I was underwhelmed by the discussion as no cohesive take away seemed to translate from the work that would advance drug development or clearly improve stratification of patients in clinics.

Major:

- 1) Given the disease etiology and progression, please discuss the relevance/ interpretability of the 5-year risk and an 11-year risk models used to model OA.
- 2) Please, refer to the study design as a retrospective case control study. Please, make changes throughout the manuscript and the labeling of Figure 1A to improve clarity of the methodology.
- 3) "An equal number of control participants who never developed OA were identified (N = 67,772). Controls were randomly selected and date-matched with the OA diagnosis dates for case patients." Why were controls not also age and sex matched as these are fixed risk factors known to influence OA?
- 4) What exclusion criteria were applied to controls? Many EHR codes for OA are common that are non-specific. Discuss the role of contamination of OA subjects in control group to interpretability of the findings.
- 5) Please provide more details on how genetic data was used in the manuscript.
 - a. For representation ii) the whole genome polygenic risk score (WGPRS) is this were PRSice was used? Or did the authors use the variant list from the GRS by Boer et al to calculate GRS for each individual. It is not clear if the authors used PRSice themselves to generate a GRS or just looked up the variants selected in the Boer manuscript for this or ever something else. If PRSice was run for this manuscript, please describe the testing and validation more clearly in terms of the cohort. A flow chart would be helpful.
- 6) Was multi-omic and biomarker data available for all participants? Or were exclusion based on this data being missing for otherwise UKB participants who would have been cases and/ or controls?
- 7) For Figure 1C, please also make this an upset plot to make clear participants may have OA at more than one site.
- 8) Define primary and secondary EHR data
- 9) "103,086 patients with OA were identified from primary and secondary EHR data14 (~21% of all UK biobank study participants, Sup File 1)." Is this supposed to be referencing Supplementary Figure 1? What I think is Sup File 1 is a spreadsheet with EHR codes and does not clearly support how sample size got to 103,086. Neither Sup File 1 or Sup Fig 1 demonstrates where the number 21% came from.
- 10) What is difference between "read_3" and "read_2" spreadsheets which I believe are Sup File 1?
- 11) Figure 4 legend needs more detail. The word OA is not even listed in the legend. Average prediction probability for any OA? Provide more detail such a reader would be able to look at the figure and legend alone and interpret findings.

12) Do the ClinGRS, ClinPath, ClinMet and/ or ClinPro provide insight to different reasons BMI increases the risk of OA? The multi-omics data has the potential to identify subgroups of high risk OA patients with high SHAP values. This is a major missed opportunity to use the multi-omics data to understand the role of BMI to OA and possibly identify subsets of patients who would benefit from different interventions.

13) Please, present the ClinGRS, ClinMet and CliPro OA risk models using the SHAP values similar to how Clin model was presented in Figure 2D? The sparse volcano plots (Figure 6B) are not particularly informative.

Minor:

1) I had to guess the order of the supplementary files. Please, edit the supplementary files to include on the first sheet the name of the file as this seems to have gotten lost in the uploading process.

2) The writing of the introduction needs polishing with paragraphs chopped into two that should be together. The introduction overall reads like a laundry list of points without stitching the gaps in the field that serve as a foundation for the valuable current work. The information is there but needs polishing.

3) Abstract: "14 sub-groups of OA risk profiles were identified, and validated in an independent set of patients evaluating the 11-year OA risk, with 88% of patients uniquely assigned to one of the sub-groups." Is a confusing sentence. Please break it up into two sentences to make findings easier to digest.

4) "The average annual cost of OA for an individual is estimated to be between \$700-\$15,600 (2019)¹." Please, specify in which countries.

5) Figure 2 legend typo "steoroid"

Reviewer #2 (Remarks to the Author):

I would like to commend the authors for their comprehensive study, which involved a vast cohort of approximately 20,000 osteoarthritis (OA) patients and matched controls from the United Kingdom. Employing interpretable machine learning techniques, the researchers successfully mapped a wide range of OA risk biomarkers, both clinical and molecular, thereby offering valuable insights into the complexities of OA pathogenesis. Their efforts led to the identification of distinct OA risk sub-groups characterized by unique biomarker profiles, significantly advancing our understanding of patient-specific OA trajectories. The model's robustness in accurately predicting 70% of OA cases is noteworthy, with key predictors including age, BMI, NSAID prescription history, vitamin D levels, and socioeconomic status. Additionally, the integration of omics data enriched risk estimations by highlighting molecular biomarkers such as CRTAC1 and COL9A1. However, my review brings to light several concerns that merit attention for the paper's improvement.

1. Terminology and Study Design:

The primary concern raised by this paper pertains to its usage of terminology. The authors assert the construction of a model predicting the five-year risk of OA, which may mislead readers to assume predictions are solely based on data available at the baseline. However, these models incorporate data gathered over the subsequent five years (or until OA diagnosis, if earlier), effectively conditioning on future observations. Furthermore, these models cannot be employed to stratify subjects into risk categories in a clinical setting since one cannot anticipate factors like the subject's use of non-steroidal anti-inflammatory drugs four years into the future, which could influence their risk of receiving an OA diagnosis five years hence. Hence, the study essentially assumes the form of a retrospective study, characterizing OA patients in the five years preceding their formal diagnosis. It is imperative that the authors either revise the paper to elucidate this approach clearly to the reader or consider redesigning their study to restrict predictions to data available at each timepoint.

2. Inclusion of Gender as a Predictor:

Gender, a well-recognized risk factor for OA, notably did not feature among the top-ranked predictors in the model. This observation warrants a thorough statistical investigation and

subsequent discussion to elucidate the underlying reasons behind this omission. In relation to that, one can for example speculate that the inclusion of vitamin D supplements serves as a proxy for women after menopause that are known to be at higher risk for OA.

3. Baseline Comparison and Model Comparison:

On page 5, line 88 Authors state: „The cases and controls were well-matched based on comparison of baseline distribution for known OA risk biomarkers (Fig. 1B)“. This claim is false as the OA cases were older, with higher BMI and higher female ratio than the controls, as expected in agreement with their associated risk of OA. Prior studies, also based on the UK Biobank cohort, have demonstrated that models solely incorporating age, sex, and BMI can predict the incidence of Hip OA with an AUC of 0.685 and Knee OA with an AUC of 0.701. It is essential to compare the authors' models with simpler models incorporating established risk factors such as age, sex, and BMI, or alternatively, consider matching controls with cases based on these parameters for a more informative analysis.

4. Data Preparation and Highly Correlated Variables:

The data preparation process needs refinement. It is for example unclear why there are more controls than cases included in the study population after filtering on OA diagnosis/matched index date less than five years from the baseline visit. Additionally, the inclusion of highly correlated variables, such as the ratios of fat mass to fat-free mass in the right leg and left leg should be reconsidered to avoid multicollinearity issues. The same principle should be applied to the ratio of fat mass to fat-free mass in the right arm vs. the left arm, as well as grip strength in the right hand vs. the left hand. Furthermore, OA cases and controls should be matched on the date of the baseline visit in the UK Biobank study to reduce potential bias.

5. Distinction between OA Presence and Diagnosis:

The paper would benefit from a clearer distinction between individuals having OA and those who have received an OA diagnosis, as OA typically presents gradually. The significance of NSAID prescriptions up to one year before OA diagnosis as a model feature underscores this nuanced aspect. NSAIDs are commonly prescribed for OA management and should preferably not be incorporated into a model designed to predict the onset of OA.

6. Causality and Interpretation:

Lastly, the authors should exercise caution in their claims of causality, as biomarkers may reflect processes already in motion and may not necessarily indicate a causal relationship. It is essential to emphasize that modifying these biomarkers does not guarantee a reduced risk of developing the disease.

Reviewer #3 (Remarks to the Author):

Nielsen and colleagues present in their paper entitled "Data-driven identification of predictive risk biomarkers for subgroups of osteoarthritis using an interpretable machine learning framework: a UK biobank study" a large study aimed at predicting individuals who might develop osteoarthritis. It is very good study which uses novel approaches to derive a prediction model for OA although the results are disappointing these findings are important to the field.

There are several parts of this study that indicate it is of high quality. Firstly, the authors use a cross-validation approach to their machine learning approach. Secondly, the well-constructed matched controls they use is a good use of the data. Finally, the breadth of variables they include is a good use of UK Biobank which is almost unique in its size.

My main criticisms/comments are as follows:

- The authors acknowledge OA is a heterogeneous condition in the introduction. Yet the majority of their study participants have an undifferentiated diagnosis of OA. We know there are considerable differences between joints for OA. For example, increased BMI is a much bigger risk factor for knee than hip OA. Obviously, from grouping all the OA diagnoses together the authors gain a degree of power but I also feel they introduce a lot of noise. I think this should be acknowledged in the discussion.

- Despite the breadth of the data they include their AUC ~ 0.7 seems to be in line with previous prediction models for OA (correct me if I am wrong). I wonder if this is because of my previous point. I think this should be discussed in the discussion.
- The follow up nature of their study is limited as only 5 years. UK Biobank has released data to 2022. Is there a reason why the authors chose only to study a 5 year period over a longer term?
- Finally, although they validate their model in a separate sample in UKB really this needs to be done in a separate cohort. I think this should be acknowledged.

Minor comments:

Line 479 – It should read UK Biobank not UK BioBank

Response to reviewer comments on manuscript "Data-driven identification of predictive risk biomarkers for subgroups of osteoarthritis using an interpretable machine learning framework: a UK biobank study" to Nature Communications.

REVIEWER COMMENTS

Reviewer #1 (Remarks to the Author):

Summary:

Osteoarthritis is a common, debilitating disease with no approved treatments for preventing disease onset or slowing progression. Most patients are identified in end-stages where the only option is joint replacement surgery. It is well known that OA is heterogeneous disease and using machine learning approach to refine the disease into subtypes may provide clarity needed to develop personalized therapies. To address this highly relevant and important problem, the authors for the current manuscript "developed a machine learning model to predict individual five-year risk of OA and identify risk biomarkers". A retrospective case control design was harnessed to identify OA cases and controls with appropriate longitudinal data. They also performed an interesting subgroup analysis identifying 14 sub groups which were validated in an independent set of patients. I found this to be a very interesting analysis overall but do have some significant comments and questions. A major missed opportunity was that the multi-omics data was not used to disentangle the role of BMI in OA. High BMI is an established predictor of OA but the underlying reasons BMI influences OA etiology and progression are likely diverse. Some patients may benefit from classic weight reduction and others from medication to reduce inflammation or even some other unknown pathways the field has not yet attributed to BMI role in OA. Further, the findings from multi-omics and clinical prediction models overall were not integrated in such a way to advance the field. Overall, I was underwhelmed by the discussion as no cohesive take away seemed to translate from the work that would advance drug development or clearly improve stratification of patients in clinics.

Major:

1) Given the disease etiology and progression, please discuss the relevance/ interpretability of the 5-year risk and an 11-year risk models used to model OA.

Thank you for highlighting many relevant points in this question. We have divided this into three response categories where we have addressed these concerns.

1. 5-year risk period

The 5-year risk period was chosen to capture patients in the time-period immediately preceding an OA diagnosis, when they are at their highest risk and preventative opportunities can be identified. We have added this paragraph to results (section: OA study populations) as to why a five year study period was chosen: *"We focussed our study on the diagnosis of OA up to five years after the assessment centre. This was to capture the risk biomarkers that are predictive of OA diagnosis in the focussed period of five years prior to diagnosis, when patients are at high-risk, and a potential window to explore for preventative interventions with the deep phenotyping of the aging population."*

The retrospective study integrates multi-modal patient data in this highly relevant 5-year period to capture the broad risk landscape in a real-world setting. The following has been added to discussion to highlight the five-year model's relevance and interpretation:

“A strength of this retrospective design is the use of all available data five years prior to OA development. This contributes to our understanding of potential windows for preventative interventions, identifying risk biomarkers when patients are at high-risk.”

“The presented models predict an individual’s five-year risk of an OA diagnosis from EHR data which encompasses the range of OA heterogeneity and pathophysiology in real-life clinical settings. The complexity of OA risk was captured using multi-modal clinical data, biochemical and molecular signatures of OA.”

2. 11-year risk period

We further identified sub-groups with distinct risk biomarker profiles from the five-year risk model. To investigate the robustness of these sub-groups and the longitudinal relevance, they were validated in an independent set of patients evaluating the 11-year OA risk as the longest data-rich period relevant to incident OA diagnoses of the UK Biobank assessment centre used in this study. The interpretation of this validation is the establishment of the relevance of the risk profiles and risk biomarkers in a longer time-period of 11-years prior to OA diagnosis, potentially when patients are at an earlier stage of disease risk. This identifies potential early opportunities for pre-emptive intervention.

The following has been added in the discussion: *“A high consistency in the sub-groups was observed across the five-year and 11-year risk populations, indicating that the major risk biomarkers, and associated risk profiles, remained relevant for the prediction of OA diagnosis across these time periods. Therefore, interventions addressing the modifiable risk factors may have an impact on OA risk, although this requires further testing.”*

3. Complex OA disease aetiology and progression

OA is a heterogeneous and progressive disease, with patients facing clinical inertia and delayed clinical diagnosis. Hence, there is an unmet need to explore and understand the risk biomarkers of progression to disease diagnosis. The machine learning approach helped to identify diverse risk biomarkers enabling an understanding of potential subpopulations at high-risk of progression to OA, by capturing the heterogeneous pathophysiology underlying OA. Here, we ranked the most predictive risk biomarkers of an OA diagnosis. The presented machine learning models provide insights to already established OA risk biomarkers as well as novel OA biomarkers, that have not previously been integrated in one combined clinical model. The combination of these established and novel risk biomarkers may contribute to the establishment of prevention strategies for OA. We discuss the context and value of risk biomarkers, in the context of early intervention, thoroughly in the discussion section.

2) Please, refer to the study design as a retrospective case control study. Please, make changes throughout the manuscript and the labeling of Figure 1A to improve clarity of the methodology.

Thank you for this comment, we fully agree and have updated this throughout the manuscript including labelling of Figure 1A.

3) “An equal number of control participants who never developed OA were identified (N = 67,772). Controls were randomly selected and date-matched with the OA diagnosis dates for case patients.” Why were controls not also age and sex matched as these are fixed risk factors known to influence OA?

It is our view that the use of the XGBoost model effectively eliminates the need for age-/sex-matching. We would argue that utilising the model architecture to remove the effect of age and sex internally, enables a better separation of the patients, as the model can use these as parameters alongside other variables. Using any risk biomarker (of known or unknown effect size) as a parameter in the model allows the model to potentially identify higher-order associations between the biomarkers, which matching would not allow the

model to capture. Additionally, this facilitates a comparison of magnitude of the feature importance of sex and age in relation to the other parameters included in the model.

Approaching sex and age as any other input biomarkers in the model has been done in multiple other studies, as also described in an opinion published in Nature Reviews Rheumatology in 2018 (Jamshidi et al., 2018 – section “Developing OA prediction models” -> “Variables in prediction models”. Link: <https://www.nature.com/articles/s41584-018-0130-5>). Additionally, in a recent review, 26 models for OA prediction were identified of which 22 used age and 16 used sex as predictors in the model (Appleyard et al., 10.1002/acr.25035).

Finally, UK Biobank recruited study participants focusing on middle- and old-aged people with the intent to explore genetic and non-genetic determinants of disease (Sudlow et al, PLOS Med, 2015). While we selected this set of study participants at random these are still matched within an appropriate age range to be at risk of cardiometabolic disease including onset of osteoarthritis.

4) What exclusion criteria were applied to controls? Many EHR codes for OA are common that are non-specific. Discuss the role of contamination of OA subjects in control group to interpretability of the findings.

Thank you – this has been important to clarify, and we have now updated the result section “OA study populations” thoroughly as well as Supplementary Fig. 1, describing our filtering on the controls.

In brief, eligible controls for the study were those with linked primary health care data, did not have a clinical diagnosis of OA (N=399,249) and did not have a death date registered, to ensure we had maximum followup data available. This left N=153,028 individuals eligible for selection as a control. Out of these, 67,772 were randomly selected to match an index date (first OA diagnosis date of cases) to match the cohort 1:1. Final filtering of controls required the matched index date to be within five years of the UK Biobank recruitment assessment centre.

We identified UKB participants diagnosed with OA using EHR codes from both primary care (GP) and secondary care (hospitalisation) data with a bespoke list of relevant OA EHR codes (Supplementary File 1). We have added the following in the methods section “OA study and validation populations” on how OA clinical codes were generated in this study: *“This list was generated using a published list of OA read codes, a curated search for OA EHR codes in UK Biobank and reviewed by co-authors with extensive clinical experience in the UK.”*

Many codes are unspecific, in that the affected joint is not specified, as seen in the new Fig. 1B (formerly 1C). However, the use of real-world clinical data does reflect the underdiagnosis of OA, which may contaminate controls. We have extended our discussion on the potential contamination of OA subjects in the control group and its impact on model interpretation further in the discussion: *“NSAID prescription one year prior to OA diagnosis was the second most important predictor. This likely reflects the delay or clinical inertia in the diagnosis of OA, reflecting several barriers to OA management in primary care that have been described previously^{21,22}. The model was trained to predict OA diagnosis, as indicated by EHRs. To minimise the risk of contamination of OA patients in the control group we used previously published clinical codes²³, in addition to a curated search for OA codes, when defining the OA study and validation populations. However, it is likely that more severe OA cases are captured with a clinical diagnosis. Hence, control contamination is most likely due to less severe OA cases, potentially minimising the impact on model performance and outputs.”*

5) Please provide more details on how genetic data was used in the manuscript.

a. For representation ii) the whole genome polygenic risk score (WGPRS) is this were PRSice was used? Or did the authors use the variant list from the GRS by Boer et al to calculate GRS for each individual. It is not clear if the authors used PRSice themselves to generate a GRS or just looked up the variants selected in the Boer

manuscript for this or ever something else. If PRSice was run for this manuscript, please describe the testing and validation more clearly in terms of the cohort. A flow chart would be helpful.

A supplementary figure presenting an overview (flow-chart) of the WGPRS/GRS generation methods has been added (Sup Fig. 8) and is referred to in both the results and methods to clarify how the genetic data was used. Briefly, we used the software PRSice to generate all genetic risk scores used in our models, using weights from an OA GWAS generated and shared to us by Boer *et al* (GWAS subset that excluded the UKB samples). These details have been made clear in the Online Methods and Acknowledgements sections too.

The new supplementary figure is the following:

Supplementary Figure 8: Schematic representation of the methods for the generation of polygenic risk scores. Single-nucleotide polymorphisms (SNPs) were aggregated at the whole-genome-level, biological pathway-level (KEGG pathways) and gene-level to use as genetic features in the osteoarthritis predictive model.

6) Was multi-omic and biomarker data available for all participants? Or were exclusion based on this data being missing for otherwise UKB participants who would have been cases and/ or controls?

Multi-omics data (genetics, metabolomics, and proteomics) was not available for all participants. Multi-omics data (i.e. including all omics data layers) was available for 424 cases diagnosed with OA and 466 controls, on which, the ClinSNP, ClinWGPRS, ClinGRS, ClinPath, ClinMet and ClinPro models were trained for exploring predictive impact of OA-specific signatures. The number of study participants per model is given in Fig. 6A and a sensitivity test conducted by retraining the Clin model only for the specific subset of participants with multi-omics data available in Sup. Table 2 (which was trained on 18625 cases and 18779 for the subset with genetics; 4502 cases and 4519 controls for the subset with metabolomics; and 1723 cases and 1816 controls for the subset with proteomics) to allow comparison of performance

after excluding samples without multi-omics. The ROC-AUC was unchanged for these specific subsets of the Clin model. We have additionally added a comment specifying this further in the results: *“Furthermore, diverse omics data were integrated into separate models for individuals where this information was available (genetics (ClinGRS/ClinPath), metabolomics (ClinMet), proteomics (ClinPro), Fig 1D).”*

7) For Figure 1C, please also make this an upset plot to make clear participants may have OA at more than one site.

Fig. 1C has been updated to an upset plot in line with the reviewer instructions. Following updates in line with Nature Communication guidelines for images, this figure is now called Fig. 1B.

8) Define primary and secondary EHR data

Thank you for this comment as this specifically relates to the UK health care system which bears clarifying in the manuscript. Primary and secondary EHR data refers to linked EHR data in UK Biobank where all study participants were registered with the National Health Service (NHS). NHS UK has four categories of care providers including primary (e.g. general practice) and secondary care (e.g. hospital) (<https://digital.nhs.uk/developer/guides-and-documentation/introduction-to-healthcare-technology/the-healthcare-ecosystem>).

We have clarified the definition of primary and secondary EHR data throughout the manuscript including in results in the section *“OA study populations”* as well as in methods section *“UK Biobank”* with appropriate references to primary health care data (general practices, available data until 09/2017) and secondary health care data (hospitalisations, available data until 03/2017).

9) *“103,086 patients with OA were identified from primary and secondary EHR data¹⁴ (~21% of all UK biobank study participants, Sup File 1).”* Is this supposed to be referencing Supplementary Figure 1? What I think is Sup File 1 is a spreadsheet with EHR codes and does not clearly support how sample size got to 103,086. Neither Sup File 1 or Sup Fig 1 demonstrates where the number 21% came from.

We have clarified this in the section *“OA study populations”* with appropriate referencing to Sup File 1 (clinical EHR codes) and Sup Fig. 1 (Study inclusion criteria): *“We identified 103,086 patients with an OA diagnosis from EHR data (~21% of all UK Biobank participants, Supplementary Fig. 1). In total, 55,628 OA diagnoses were identified from primary care settings (general practices, follow-up until 09/2017) and 49,318 OA diagnoses from secondary care settings (hospital inpatient data, follow-up until 03/2017). Clinical codes of OA diagnoses are listed in Supplementary File 1 (primary care: Read v2 and CTV3/Read v3, secondary care: ICD-9 or ICD-10).”*

10) What is difference between “read_3” and “read_2” spreadsheets which I believe are Sup File 1?

Two EHR coding systems are used in UK primary care (general practice) data: Read v2 and CTV3/Read v3 codes (CTV3/Read v3 represent an updated coding system). Both are used for filtering of clinical data in this study, see Supplementary Files 1, 5 and 6 for lists of codes used. This information has been clarified in the Supplementary files (including Supplementary File 1) and in Methods.

11) Figure 4 legend needs more detail. The word OA is not even listed in the legend. Average prediction

probability for any OA? Provide more detail such a reader would be able to look at the figure and legend alone and interpret findings.

The legend has been updated to be more descriptive, from *“Cluster prediction metric, defining rules and validation in an independent population”* to *“Cluster prediction metrics and defining rules in the osteoarthritis (OA) study population (time-window between data collection at the assessment centre and OA diagnosis (any OA): less than 5 years) and validation in independent hold-out cohort (time-window: 5 to 11 years)”*

The abbreviations Avg Pred Prob and PPV are now also described in the legend.

Additionally, inspired by the reviewer’s comment, we have for all figures further included abbreviation definitions for OA and NSAIDs where appropriate OA = Osteoarthritis. NSAIDs = non-steroidal anti-inflammatory steroid drugs.

12) Do the ClinGRS, ClinPath, ClinMet and/ or ClinPro provide insight to different reasons BMI increases the risk of OA? The multi-omics data has the potential to identify subgroups of high risk OA patients with high SHAP values. This is a major missed opportunity to use the multi-omics data to understand the role of BMI to OA and possibly identify subsets of patients who would benefit from different interventions.

We thank the reviewer for the comment and agree it would indeed be substantial progress to be able to utilise multi-omics data on all participants. We were however significantly limited as the overlap in available omics layers offered low power (all data layers only available for 424 cases diagnosed with OA and 466 controls). As the omics data is assessed on more participants, there will be an opportunity to explore this further in future releases of UK Biobank.

In our present study, we have used omics data (genetics, proteomics and metabolomics) as separate data layers (on the available subset) added to the Clinical model to i) determine if they add to the overall prediction performance, compared to the Clin model; and ii) prioritise omics feature by their relative feature importance in the predictive models.

We have now added new results in Fig. 3D i.e. the proteomic profiles for biomarkers that are most differentiating OA cases in each cluster (full data available in Sup File 3). This allows us to explore molecular levels of protein biomarkers between the identified risk subgroups. Relevant text for this analysis has been added in the methods and results sections.

We further discuss the impact of the proteins in the discussion and how these relate to obesity, OA as well as other relevant pathways: *“We also explored which proteins were differentially expressed across subgroups of OA risk. These may reflect differences in relevant biological pathways such as inflammation (e.g. IL6, CXCL6), body weight regulation/energy homeostasis (e.g. LEP, GDF15), but also in proteins directly relevant to osteoarthritis, such as the proteoglycan ACAN (aggrecan)³³ and AMBP (bikunin precursor)³⁴. Furthermore, HGF, which has previously been shown to be involved in both obesity³⁵ and OA biologies³⁶, is significantly over-expressed in OA cases from most clusters with higher BMI and under-expressed in most lower BMI clusters. Therefore, despite HGF being associated with OA in osteoblasts, its plasma levels may be more heterogenous.”*

For the ClinPro model, age, NSAIDs and BMI along with CRTAC1 and COL9A1 were the top five most important features for prediction of OA. BMI SHAP values had no significant interaction with CRTAC1 and COL9A1 normalised expression levels. This suggests the importance of risk biomarkers in addition to BMI for predicting OA risk. Similar analysis was conducted for ClinGRS, ClinPath, ClinMet (results not shown).

13) Please, present the ClinGRS, ClinMet and CliPro OA risk models using the SHAP values similar to how Clin model was presented in Figure 2D? The sparse volcano plots (Figure 6B) are not particularly informative.

Figure 6B has been updated to represent SHAP plots for the ClinGRS (Fig. 6B), ClinPath (Fig. 6C), ClinMet (Fig. 6D) and ClinPro (Fig. 6E) models. Additional plots are also included in the new Supplementary Fig. 9, 11 and 12.

Minor:

1) I had to guess the order of the supplementary files. Please, edit the supplementary files to include on the first sheet the name of the file as this seems to have gotten lost in the uploading process.

We are sorry to hear this got lost during the uploading process of the xlsx files. Supplementary Files 1-6 now include the name of the file on the first sheet as well as a description of the sheets and their content. In the Supplementary File 7 PDF, this is marked in the top corner of the page. Furthermore, the content of Supplementary File 7, the TRIPOD checklist, has been updated to correct page numbers following review of the manuscript.

2) The writing of the introduction needs polishing with paragraphs chopped into two that should be together. The introduction overall reads like a laundry list of points without stitching the gaps in the field that serve as a foundation for the valuable current work. The information is there but needs polishing.

We thank the reviewer for calling this out. We have now extensively rewritten the introduction to address the reviewer comment and improve readability.

3) Abstract: "14 sub-groups of OA risk profiles were identified, and validated in an independent set of patients evaluating the 11-year OA risk, with 88% of patients uniquely assigned to one of the sub-groups." Is a confusing sentence. Please break it up into two sentences to make findings easier to digest.

The abstract sentence has now been updated to the following: "*We identified 14 sub-groups of OA risk profiles. These sub-groups were validated in an independent set of patients evaluating the 11-year OA risk, with 88% of patients being uniquely assigned to one of the 14 sub-groups.*"

4) "The average annual cost of OA for an individual is estimated to be between \$700-\$15,600 (2019)¹." Please, specify in which countries.

This spans across multiple countries and for specific details we refer the reader to the referenced manuscript by Leifer *et al*, 2022, Osteoarthritis Cartilage. We have updated the introduction to specify different continents upon which Leifer *et al* has based their data in the introduction: "*The average annual cost of OA for an individual is estimated to be between \$700-\$15,600 (USD, 2019) across countries in Asia, Europe, North America and Oceania.*"

5) Figure 2 legend typo "steoroid"

"steoroid" has been corrected to "steroid"

Reviewer #2 (Remarks to the Author):

I would like to commend the authors for their comprehensive study, which involved a vast cohort of approximately 20,000 osteoarthritis (OA) patients and matched controls from the United Kingdom. Employing interpretable machine learning techniques, the researchers successfully mapped a wide range of OA risk biomarkers, both clinical and molecular, thereby offering valuable insights into the complexities of OA pathogenesis. Their efforts led to the identification of distinct OA risk sub-groups characterized by unique biomarker profiles, significantly advancing our understanding of patient-specific OA trajectories. The model's robustness in accurately predicting 70% of OA cases is noteworthy, with key predictors including age, BMI, NSAID prescription history, vitamin D levels, and socioeconomic status. Additionally, the integration of omics data enriched risk estimations by highlighting molecular biomarkers such as CRTAC1 and COL9A1. However, my review brings to light several concerns that merit attention for the paper's improvement.

1. Terminology and Study Design:

The primary concern raised by this paper pertains to its usage of terminology. The authors assert the construction of a model predicting the five-year risk of OA, which may mislead readers to assume predictions are solely based on data available at the baseline. However, these models incorporate data gathered over the subsequent five years (or until OA diagnosis, if earlier), effectively conditioning on future observations. Furthermore, these models cannot be employed to stratify subjects into risk categories in a clinical setting since one cannot anticipate factors like the subject's use of non-steroidal anti-inflammatory drugs four years into the future, which could influence their risk of receiving an OA diagnosis five years hence. Hence, the study essentially assumes the form of a retrospective study, characterizing OA patients in the five years preceding their formal diagnosis. It is imperative that the authors either revise the paper to elucidate this approach clearly to the reader or consider redesigning their study to restrict predictions to data available at each timepoint.

Thank you for highlighting this and we agree that retrospective data-capture has been used to evaluate the risk of an OA diagnosis. The text has been updated throughout to reflect that this is a retrospective study using machine learning to predict the risk of an OA diagnosis.

2. Inclusion of Gender as a Predictor:

Gender, a well-recognized risk factor for OA, notably did not feature among the top-ranked predictors in the model. This observation warrants a thorough statistical investigation and subsequent discussion to elucidate the underlying reasons behind this omission. In relation to that, one can for example speculate that the inclusion of vitamin D supplements serves as a proxy for women after menopause that are known to be at higher risk for OA.

We thank the reviewer for highlighting this aspect as it warrants some discussion and the useful insight on vit D supplements as a possible proxy. In order to address this, we have added the following analysis to results: *"A post-hoc analysis showed that individuals that reported taking vitamin D supplements typically had higher levels of vitamin D, a higher age and was more common in women with menopause which might contribute to the observed association between higher vitamin D levels and OA risk (Supplementary Fig. 4A, 4B, and 4C respectively)"*.

Furthermore, we added this paragraph to the discussion "Sex is an established risk factor for OA, with women being at higher risk of developing OA compared to men¹. In our Clin model, sex only modestly contributed to prediction of OA, being ranked the 85th most predictive feature. This is likely because the Clin model included

other features that potentially reflect sex as an OA risk factor. Sex was strongly correlated ($R^2 > 0.75$) with testosterone (ranked 35th) as well as all measurements of body composition metrics of fat mass to fat free mass (ranked between 24–60th). Based on SHAP, lower total testosterone and higher fat mass/fat-free mass ratio were associated with higher risk of OA and female sex. It is also possible that sex was not highly ranked due to the complex relationship with vitamin D levels. In this study, higher vitamin D levels were associated with higher risk of OA. Our analysis showed that individuals that reported taking vitamin D supplements were older and had higher levels of vitamin D. Additionally, a higher proportion of individuals taking vitamin D supplements was observed amongst post-menopausal women (a known high-risk group for OA^{26,27}), compared to men or pre-menopausal women. Therefore, vitamin D levels in the Clin model may be capturing an older population and post-menopausal women who take vitamin supplements. These findings suggest that other parameters in the model account for not identifying sex as highly ranked in our model”.

3. Baseline Comparison and Model Comparison:

On page 5, line 88 Authors state: „The cases and controls were well-matched based on comparison of baseline distribution for known OA risk biomarkers (Fig. 1B)“. This claim is false as the OA cases were older, with higher BMI and higher female ratio than the controls, as expected in agreement with their associated risk of OA.

Prior studies, also based on the UK Biobank cohort, have demonstrated that models solely incorporating age, sex, and BMI can predict the incidence of Hip OA with an AUC of 0.685 and Knee OA with an AUC of 0.701. It is essential to compare the authors' models with simpler models incorporating established risk factors such as age, sex, and BMI, or alternatively, consider matching controls with cases based on these parameters for a more informative analysis.

We thank the reviewer for correcting this and suggesting a comparison with simpler models. We have now corrected results presenting the OA case and control arm comparison of baseline distribution in the Results section “OA study population” to: *“Comparison of baseline distribution for known OA risk biomarkers showed that OA cases in general were of older age, with higher BMI and a higher female-to-male ratio in comparison to controls in both the OA study population and hold-out validation population (Fig. 1C)”*.

We have included new results using a simpler model using only these known risk factors. On their own, age, sex and BMI showed modest predictive performance (Age-only model ROC-AUC: 0.64 (0.63 – 0.64), BMI-only model ROC-AUC: 0.59 (0.58 – 0.60), Sex-only model: 0.53 (0.52 – 0.53)). When using all three features on their own, the model performs with ROC-AUC: 0.67 (0.67 – 0.68). The following has been added to results: *“A baseline ML model predicting the five-year risk of OA, using only age, sex, and BMI, that are well-known OA risk biomarkers, predicted OA with ROC-AUC: 0.67 (95% CI: 0.67 – 0.68). Comparison of the predicted risk probabilities at the individual level showed that the additional features in the Clin model compared to the simpler model (age, sex and BMI) significantly increased the confidence in individual risk predictions (Supplementary Fig. 3) in addition to increased predictive performance.”*.

Additional analyses have also been reflected in the discussion:

“The Clin model demonstrated strong performance, aligned with previously published models¹⁴. Previous models that reached above ROC-AUC 0.75 usually included imaging variables (e.g. X-rays), pain scores, or data regarding osteoporosis and previous leg injury^{15–17}. These variables were not available in our model and might impact real-life applicability, as this data may not be widely available in primary care. The comparison to other published OA models is limited by differences in prediction horizons, age and sex distributions, and OA definitions. Our model considered a diverse OA phenotype identified from EHR codes across patients

diagnosed with OA, across mostly undifferentiated joints specifications. Furthermore, it quantified the predictive impact for a range of OA risk biomarkers, including genetics, clinical biomarkers and environmental factors, to which previous studies have provided limited insights^{4,8,9}. Some of these risk biomarkers are recognised as OA risk biomarkers (e.g. age and BMI)¹¹, while others are not traditionally considered OA risk biomarkers (e.g. personal health rating, hand grip strength, body composition, and walking pace). This highlights the novelty of the OA risk models in our study, offering new opportunities for prevention by addressing novel modifiable risk factors. These biomarkers improved the predictive performance and confidence across individual-level predictions, compared to a simpler model based only on well-known OA risk biomarkers (age, sex and BMI)."

4. Data Preparation and Highly Correlated Variables:

The data preparation process needs refinement. It is for example unclear why there are more controls than cases included in the study population after filtering on OA diagnosis/matched index date less than five years from the baseline visit.

Additionally, the inclusion of highly correlated variables, such as the ratios of fat mass to fat-free mass in the right leg and left leg should be reconsidered to avoid multicollinearity issues. The same principle should be applied to the ratio of fat mass to fat-free mass in the right arm vs. the left arm, as well as grip strength in the right hand vs. the left hand.

Furthermore, OA cases and controls should be matched on the date of the baseline visit in the UK Biobank study to reduce potential bias.

We thank the reviewer for the elaborate comment and have divided this into three response categories where we have addressed these concerns.

1. Clarification regarding Data preparation of OA cases and controls:

This has been updated in the result section "OA study populations" as well as updated Supplementary Fig. 1 (study inclusion selection overview). The reason why there are fewer cases than controls is solely based on data availability for individuals in relation to their OA diagnosis/matched index date (controls) being available within five years after the first UK Biobank assessment centre visit (study period: 2006-2010).

2. Clarification regarding multicollinearity issues:

The machine learning approach that we applied in the model by itself accounts for multi-collinearity. For modelling OA risk, a XGBoost decision tree-based classifier has been used. By nature of decision trees, each feature is evaluated for prediction of the outcome. The XGBoost models are tuned to randomly sample for both i) subsampling of individuals included to train the model in every boosting iteration (80% uniform sampling) and ii) subsampling of available features (80%). This strategy serves two purposes; i) reduce risk of overfitting ii) learn from the different insights across potentially correlated features as these may be selected together or separately to build the tree and hence help handle issues of multicollinearity. The uniform sample method ensures each training data has a similar probability of being selected.

As a sensitivity check to address potential linear multicollinearity issues, we retrained the Clin model following removal of highly correlated features (absolute Pearson correlation > 0.75) which resulted in ROC-AUC 0.72 (similar to the original model) as well as similar ranking of features based on the SHAP feature importance. We have not included these results in the manuscript, as the XGBoost model is able to pick up on non-linear interactions that we cannot account for.

3. Bias regarding baseline visit:

This is a retrospective longitudinal study design where controls are matched in relation to the OA diagnosis. We have added the following considerations of the study design to the discussion: *“We present a retrospective case-control study that extracted five years of longitudinal data prior to OA diagnosis, or control index date, for risk modelling. A strength of this retrospective design is the use of all available data five years prior to OA development. This contributes to our understanding of potential windows for preventative interventions, identifying risk biomarkers when patients are at high-risk. The Clin model demonstrated strong and robust predictive performance, with a performance aligned with previously published models¹⁴.”*

Conversely, if our study had been based on prevalent OA diagnosis at the time of the recruitment assessment centre (baseline visit), many incident cases of OA would not have been captured. This would have resulted in a reduction of power to explore the predictive impact of longitudinal risk biomarkers in our machine learning approach.

5. Distinction between OA Presence and Diagnosis:

The paper would benefit from a clearer distinction between individuals having OA and those who have received an OA diagnosis, as OA typically presents gradually. The significance of NSAID prescriptions up to one year before OA diagnosis as a model feature underscores this nuanced aspect. NSAIDs are commonly prescribed for OA management and should preferably not be incorporated into a model designed to predict the onset of OA.

We agree. The models in this study were designed to predict the clinical diagnosis of OA, as indicated by primary care (GP) and secondary care (hospitalisation) data. The models do not predict the initiation of OA disease. The manuscript text has been updated throughout to specify that the models predict the diagnosis of OA. The improved understanding of risk biomarkers for OA diagnosis will hopefully result in the earlier and more extensive diagnosis of OA.

On the prescription of NSAIDs, please see the following update in the discussion: *“NSAID prescription one year prior to OA diagnosis was the second most important predictor. This likely reflects the delay or clinical inertia in the diagnosis of OA, reflecting several barriers to OA management in primary care that have been described previously^{21,22}. The model was trained to predict OA diagnosis, as indicated by EHRs. To minimise the risk of contamination of OA patients in the control group we used previously published clinical codes²³, in addition to a curated search for OA codes, when defining the OA study and validation populations. However, it is likely that more severe OA cases are captured with a clinical diagnosis. Hence, control contamination is most likely due to less severe OA cases, potentially minimising the impact on model performance and outputs.”*

6. Causality and Interpretation:

Lastly, the authors should exercise caution in their claims of causality, as biomarkers may reflect processes already in motion and may not necessarily indicate a causal relationship. It is essential to emphasize that modifying these biomarkers does not guarantee a reduced risk of developing the disease.

The text throughout the manuscript has been updated to emphasize that our predictive model cannot be used to assess whether the identified risk biomarkers are causal of OA. It can only be concluded that these risk biomarkers were predictive of an OA diagnosis prior to this event. The following has been added to the discussion: *“Our study cannot assess whether intervention on risk biomarkers would decrease OA risk, the extent of this decrease, or the causal relationship between the risk biomarkers and OA.”*

Reviewer #3 (Remarks to the Author):

Nielsen and colleagues present in their paper entitled “Data-driven identification of predictive risk biomarkers for subgroups of osteoarthritis using an interpretable machine learning framework: a UK biobank study” a large study aimed at predicting individuals who might develop osteoarthritis. It is very good study which uses novel approaches to derive a prediction model for OA although the results are disappointing these findings are important to the field.

There are several parts of this study that indicate it is of high quality. Firstly, the authors use a cross-validation approach to their machine learning approach. Secondly, the well-constructed matched controls they use is a good use of the data. Finally, the breadth of variables they include is a good use of UK Biobank which is almost unique in its size.

My main criticisms/comments are as follows:

- The authors acknowledge OA is a heterogeneous condition in the introduction. Yet the majority of their study participants have an undifferentiated diagnosis of OA. We know there are considerable differences between joints for OA. For example, increased BMI is a much bigger risk factor for knee than hip OA. Obviously, from grouping all the OA diagnoses together the authors gain a degree of power but I also feel they introduce a lot of noise. I think this should be acknowledged in the discussion.

The diagnosis of OA was based on codes and most of the codes found were not joint specific and hence the model was trained to predict all OA. However, we felt inspired by the reviewer’s comment and have added new results from an explorative analysis on potentially different OA risk biomarkers across different joints for subsets of cases diagnosed with OA in the knee (largest subgroup, N=5341 OA cases), hip (N=2408 OA cases), arm (N=2052 OA cases), spine (N=1964 OA cases) and foot (N=758 OA cases). We fully agree and acknowledge these analyses are less well-powered, but they might provide insights to generate new hypothesis in future studies.

The new results from this analysis have been added to results as part of a new section “*OA risk biomarker heterogeneity across joints*”, as well as Fig. 7A, 7B and 7C and Supplementary Fig. 14 – 18. We found no major differences in predictive performance were observed when comparing to the overall Clin model except for the foot model. Age and prescription of NSAIDs one year ahead of the OA diagnosis were still important for prediction of OA risk stratified per joint. However, BMI had varying importance, with increased importance for predicted risk of an OA diagnosis in weight-bearing joints (knee, hip and foot) compared to arm and spine.

We discussed the new results as well as their strength and limitation in the discussion: *“In our study, most participants had an undifferentiated diagnosis of OA, potentially introducing noise in the model, reflecting heterogeneous biological mechanisms and phenotypes^{5,6}. To address this further, we explored joint-specific models and concluded, in line with previous studies, that BMI is a higher risk biomarker for OA in weight-bearing joints, among other joint-specific risk biomarker profiles. However, future studies in larger subset of patients with joint-specific effects should further validate these findings and allow better understanding of joint-specific OA pathogenesis.”*

- Despite the breadth of the data they include their AUC ~0.7 seems to be in line with previous prediction models for OA (correct me if I am wrong). I wonder if this is because of my previous point. I think this should be discussed in the discussion.

We have included a paragraph on the impact of that the majority of the study participants have an undifferentiated diagnosis and added additional comparisons with other published models in the discussion: *“The Clin model demonstrated strong and robust predictive performance, with a performance aligned with previously published models¹⁴. Previous models that reached above ROC-AUC 0.75 usually included imaging variables (e.g. X-rays), pain scores, or data regarding osteoporosis and previous leg injury as predictors and outcomes^{15–17}. These variables were not available in our model and might impact real-life applicability of the model, as this data may not be widely available in primary care. The comparison to other published OA models is limited by differences in prediction horizons, age and sex distributions, and OA definitions. Our model considered a diverse OA phenotype identified from EHR codes across patients diagnosed with OA across mostly undifferentiated joints specifications. Furthermore, it quantified the predictive impact for a range of OA risk biomarkers, including genetics, clinical biomarkers and environmental factors, to which previous studies have provided limited insights^{4,8,9}.”*

“Although the integration of omics did not improve overall model performance, it did change risk biomarker ranking. Other studies have included genetic factors, but also failed to improve model performance^{15,37}. A strength of our study is the interpretable AI, with the impact of omics highlighting relevant biological pathways and guiding OA-specific prevention strategies.”

- The follow up nature of their study is limited as only 5 years. UK Biobank has released data to 2022. Is there a reason why the authors chose only to study a 5-year period over a longer term?

Although there is hospital episode statistics (HES) diagnosis data available up until 2022 in the current release of the UK Biobank, the extract of the HES diagnosis codes used (frozen in 02/2021) only includes data until 2017. Further, the Clin model, which is central to the paper, is built upon the primary care EHR linkage data, which is only available until 09/2017 in UK Biobank (see https://biobank.ndph.ox.ac.uk/ukb/exinfo.cgi?src=Data_providers_and_dates). The cut-off dates for our data are now stated in Methods as well as results (primary care (Read v2 and CTV3/Read v3 registrations from general practices, follow-up until 09/2017) and secondary care (ICD-9 or ICD-10 registrations from hospitalisations, follow-up until 03/2017)).

We have further added this paragraph to results (section: OA study populations) as of why a five year study period was chosen: *“We focussed our study on the diagnosis of OA up to five years after the assessment centre. This was to capture the risk biomarkers that are predictive of OA diagnosis in the focussed period of five years prior to diagnosis, when patients are at high-risk, and a potential window to explore for preventative interventions with the deep phenotyping of the aging population.”*

- Finally, although they validate their model in a separate sample in UKB really this needs to be done in a separate cohort. I think this should be acknowledged.

The discussion has been updated to reflect that the external validation of these models in an independent cohort is needed: *“Finally, validation in a range of cohorts representing a diversity of genetic, cultural backgrounds and healthcare practices would further our understanding of the impact of this contextual information on OA risk.”*

Minor comments:

Line 479 – It should read UK Biobank not UK BioBank

We have corrected instances of “UK BioBank” and “UK biobank” to “UK Biobank” to reflect the proper stylization throughout the manuscript.

REVIEWERS' COMMENTS

Reviewer #1 (Remarks to the Author):

The authors have sufficiently addressed my comments. I have nothing further.

Reviewer #2 (Remarks to the Author):

Thank you for your detailed responses to my comments. I appreciate the effort you have put into addressing the concerns raised.

I am satisfied with the changes made and have no further comments to add.

Reviewer #3 (Remarks to the Author):

I have no further comments.